# Skill-Conditioned Policy Optimization with Successor Features Representations

## Abstract

A key aspect of intelligence is the ability to exhibit a wide range of behaviors to adapt to unforeseen situations. Designing artificial agents that are capable of showcasing a broad spectrum of skills is a long-standing challenge in Artificial Intelligence. In the last decade, progress in deep reinforcement learning has enabled to solve complex tasks with high-dimensional, continuous state and action spaces. However, most approaches return only one highly-specialized solution to a single problem. We introduce a **S**kill-**C**onditioned **OP**timal **A**gent (SCOPA) that leverages successor features representations to learn a continuous range of skills that solve a task. We extend the generalized policy iteration framework with a policy skill improvement update based on successor features that is analogous to the classic policy improvement update. This novel skill improvement update enables to efficiently learn executing skills. From this result, we develop an algorithm that seamlessly unifies value function and successor features policy iteration with constrained optimization to (1) maximize performance, while (2) executing the desired skills. Compared with other skill-conditioned reinforcement learning methods, SCOPA reaches significantly higher performance and skill space coverage on challenging continuous control locomotion tasks with various types of skills. We also demonstrate that the diversity of skills is useful in five downstream adaptation tasks. Videos of our results are available at: `bit.ly/scopa`.

## 1 Introduction

Reinforcement Learning (RL) has enabled groundbreaking achievements like mastering discrete games (Mnih et al., 2013; Silver et al., 2016) but also continuous control domains for locomotion (Haarnoja et al., 2019; Heess et al., 2017) and manipulation (OpenAI et al., 2019). These milestones have showcased the extraordinary potential of RL algorithms in solving specific problems.

In contrast, human intelligence is beyond mastering a single task, and adapts to unforeseen environments by combining skills. Empowering artificial agents with diverse skills was shown to improve exploration (Gehring et al., 2021), to facilitate knowledge transfer (Eysenbach et al., 2018), to enable hierarchical problem-solving (Allard et al., 2022), to enhance robustness and adaptation (Kumar et al., 2020; Cully et al., 2015) and finally, to foster creativity (Zahavy et al., 2023; Lehman et al., 2020).

Following this observation, methods have been developed to make agents more versatile, including Goal-Conditioned Reinforcement Learning (GCRL) (Liu et al., 2022), Unsupervised Reinforcement Learning (URL) (Eysenbach et al., 2018; Sharma et al., 2019), Quality-Diversity (QD) optimization (Pugh et al., 2016; Cully & Demiris, 2018; Zahavy et al., 2022), and reward design (Margolis & Agrawal, 2022). However, designing algorithms to learn large ranges of skills that are useful to solve downstream tasks remains a challenge.

In this work, we focus on learning a policy conditioned on a continuous range of pre-defined skills while also maximizing performance. Approaches like DOMiNO (Zahavy et al., 2022) or QD optimization share the objective of finding diverse and optimal skills but they only discover a finite set of solutions. Specifically, DOMiNO learns $N$ diverse policies that are different from one another, but these policies are not trained to execute pre-defined skills. Other methods like GCRL and URL try to reach goals or execute skills while disregarding other objectives like safety or efficiency, leaving a gap in our quest for machines that can execute expressive and optimal skills to solve complex tasks.

We introduce a Skill-Conditioned Optimal Agent (SCOPA) that leverages successor features representations to learn a continuous range of expressive skills while maximizing performance. Our contributions are as follows: First, we derive a policy skill improvement update analogous to the classic policy (performance) improvement update. Second, we use the policy skill improvement update to develop a theoretically justified algorithm that combines successor features with universal function approximators to learn an infinite range of expressive skills efficiently. Third, we seamlessly unify value function and successor features policy iteration with constrained optimization to (1) maximize performance, while (2) executing desired skills.

We evaluate our approach on multiple challenging continuous control tasks combined with four different skills and show that SCOPA manages to execute more skills and achieves higher performance than other baselines. Finally, we show that the skills can be used to adapt to downstream tasks in a few shots or through hierarchical learning.

## 2 BACKGROUND

### 2.1 REINFORCEMENT LEARNING

We consider the standard reinforcement learning framework (Sutton & Barto, 2018) where an agent sequentially interacts with a *Markov Decision Process* (MDP) in order to maximize the expected sum of rewards. At each time step $t$, the agent observes a *state* $s_t \in \mathcal{S}$ and takes an *action* $a_t \in \mathcal{A}$, which causes the environment to transition to a next state $s_{t+1} \in \mathcal{S}$ and to give a reward $r_t \in \mathbb{R}$, sampled from the dynamics $p(s_{t+1}, r_t \mid s_t, a_t)$. We denote $\rho^\pi(s) = \lim_{t \to \infty} P(s_t = s | s_0, \pi)$ the stationary distribution of states under $\pi$, which we assume exists and is independent of $s_0$ (Sutton et al., 1999). Additionally, we focus on the case where each state $s_t$ is associated with observable features $\phi_t = \phi(s_t) \in \Phi \subset \mathbb{R}^d$. In other cases, $\phi$ can be learned with a neural network (Grillotti & Cully, 2022b), but that is not the focus of the present work.

The objective of the agent is to find a policy $\pi$ that maximizes the expected discounted sum of rewards, or expected return $\mathbb{E}_\pi \left[ \sum_t \gamma^t r_t \right]$. The so-called value-based method in RL rely on the concept of *value function* $v^\pi(s)$, defined as the expected return obtained when starting from state $s$ and following policy $\pi$ thereafter (Puterman, 1994): $v^\pi(s) = \mathbb{E}_\pi \left[ \sum_{i=0}^\infty \gamma^i r_{t+i} \middle| s_t = s \right]$.

In addition to the value function, we also leverage the concept of *successor features* $\psi^\pi(s)$, which is the expected discounted sum of features obtained when starting from state $s$ and following policy $\pi$ thereafter (Barreto et al., 2017): $\psi^\pi(s) = \mathbb{E}_\pi \left[ \sum_{i=0}^\infty \gamma^i \phi_{t+i} \middle| s_t = s \right]$. The successor features captures the expected feature occupancy under a given policy, offering insights into the agent's future behavior and satisfies a Bellman equation in which $\phi_t$ plays the role of the reward $\psi^\pi(s) = \phi_t + \gamma \mathbb{E}_\pi \left[ \psi^\pi(s_{t+1}) | s_t = s \right]$, and can be learned with any RL methods (Dayan, 1993).

In practice, we make use of a universal value function approximator $v^\pi(s, \mathbf{z})$ (Schaul et al., 2015) and of a universal successor features approximator $\psi^\pi(s, \mathbf{z})$ (Borsa et al., 2018) that depend on state $s$ but also on the skill $\mathbf{z}$ conditioning the policy. The value function quantifies the performance while the successor features characterizes the behavior of the agent, both starting from state $s$ and conditioned on skill $\mathbf{z}$. For conciseness, we omit $\pi$ from the notations $\rho^\pi$, $v^\pi$ and $\psi^\pi$ in the following sections.

### 2.2 WORLD MODELS

Learning a skill-conditioned function approximator is challenging because in general, the agent will only see a small subset of possible $(s, \mathbf{z})$ combinations (Schaul et al., 2015; Borsa et al., 2018). In that case, a world model can be used to improve sample efficiency. One key advantage of model-based methods is to learn a compressed spatial and temporal representation of the environment to train a simple policy that can solve the required task (Ha & Schmidhuber, 2018). World models are particularly valuable for conducting simulated rollouts in imagination which can subsequently inform the optimization of the agent's behavior, effectively reducing the number of environment interactions required for learning (Hafner et al., 2019a). Moreover, world models enable to compute straight-through gradients, which backpropagate directly through the learned dynamics (Hafner et al., 2023). Most importantly, the small memory footprint of imagined rollouts enables to sample thousands of on-policy trajectories in parallel (Hafner et al., 2023), making possible to learn skill-conditioned function approximators with massive skill sampling in imagination.

In this work, we use a Recurrent State Space Model (RSSM) from Hafner et al. (2019b). At each iteration, the world model $\mathcal{W}$ is trained to learn the transition dynamics, and to predict the observation, reward, and termination condition. An *Imagination* MDP $(\widetilde{\mathcal{S}}, \mathcal{A}, \widehat{p}, \gamma)$, can then be defined from the latent states $\widetilde{s} \in \widetilde{\mathcal{S}}$ and from the dynamics $\widehat{p}$ of $\mathcal{W}$. In parallel, DREAMERV3 trains a critic network $\widehat{v}(\widetilde{s}_t)$ to regress the $\lambda$-return $V_\lambda(\widetilde{s}_t)$ (Sutton & Barto, 2018). Then, the actor is trained to maximize $V_\lambda$, with an entropy regularization for exploration: $J(\theta_\pi) = \mathbb{E}_{\substack{a \sim \pi(.|\widetilde{s}) \\ \widetilde{s}' \sim \widehat{p}(.|\widetilde{s}, a)}} \left[ \sum_{t=1}^{H} V_\lambda(\widetilde{s}_t) \right]$.

## 3 PROBLEM STATEMENT

In this work, we define a skill $\mathbf{z} \in \mathcal{Z}$ executed by a policy as the expected features under the policy's stationary distribution, $\mathbf{z} = \lim_{T \to \infty} \frac{1}{T} \sum_{t=0}^{T-1} \boldsymbol{\phi}_t = \mathbb{E}_{s \sim \rho} [\boldsymbol{\phi}(s)]$, that we simply note $\mathbb{E}_\pi [\boldsymbol{\phi}(s)]$. Given this definition, we intend to learn a skill-conditioned policy $\pi(\cdot|\cdot, \mathbf{z})$ that (1) maximizes the expected return, and (2) is subject to the expected features under the policy $\pi(\cdot|\cdot, \mathbf{z})$ converges to the desired skill $\mathbf{z}$, which corresponds to the following constrained optimization problem:

$$\forall \mathbf{z} \in \mathcal{Z}, \quad \text{maximize } \mathbb{E}_{\pi(\cdot|\cdot, \mathbf{z})} \left[ \sum_{i=0}^{\infty} \gamma^i r_{t+i} \right] \quad \text{subject to } \mathbb{E}_{\pi(\cdot|\cdot, \mathbf{z})} [\boldsymbol{\phi}(s)] = \mathbf{z} \quad \text{(P1)}$$

For example, consider a robot whose objective is to minimize energy consumption, and where the features characterize the velocity of the robot $\boldsymbol{\phi}_t = \mathbf{v}_t = [v_x(t) \quad v_y(t)]^\mathsf{T}$ and the skill space $\mathcal{Z} = [-v_{\max}, v_{\max}]^2$. For each desired velocity $\mathbf{z} \in \mathcal{Z}$, $\pi(\cdot|\cdot, \mathbf{z})$ is expected to (1) minimize energy consumption, while (2) following the desired velocity $\mathbf{z}$ in average, $\lim_{T \to \infty} \frac{1}{T} \sum_{t=0}^{T-1} \mathbf{v}_t = \mathbf{z}$.

Now consider another example, where the objective is to go forward as fast as possible, and the features characterize which foot is in contact with the ground at each time step. For example, $\boldsymbol{\phi}_t = [1 \quad 0]^\mathsf{T}$ for a biped robot that is standing on its first leg and with the second leg not touching the ground at time step $t$. With these features, the $i$-th component of the skill $\mathbf{z}$ will be the proportion of time during which the $i$-th foot of the robot is in contact with the ground over a trajectory, denoted as feet contact rate. In that case, the skill space is continuous and characterizes the myriad of ways the robot can walk and specifically, how often each leg is being used. In comparison with the skill space, the feature space is finite and contains only four elements, $\Phi = \{0, 1\}^2 \subset \mathcal{Z} = [0, 1]^2$. In order to achieve a feet contact of $\mathbf{z} = [0.1 \quad 0.6]^\mathsf{T}$, the robot needs to use 10% of the time the first foot and 60% of the time the second foot, and necessarily needs multiple time steps to execute the skill. Notice that in that case, the feature space is smaller than the skill space $\Phi \subsetneq \mathcal{Z}$.

Problem P1 amounts to maximizing the return *while* executing a desired skill. If we only consider the constraint, we can show that our problem is related to GCRL. In GCRL, a common assumption is to consider that achieving a desired goal $\mathbf{g}$ is equivalent to reaching a state that is in the subset $\{s \in \mathcal{S} | \boldsymbol{\phi}(s) = \mathbf{g}\}$ (Liu et al., 2022). In contrast, our constraint is to execute a desired skill $\mathbf{z}$, i.e. to sample a trajectory that is in the subset $\{(s_t)_{t \geq 0} \in \mathcal{S}^\mathbb{N} | \lim_{T \to \infty} \frac{1}{T} \sum_{t=0}^{T-1} \boldsymbol{\phi}(s_t) = \mathbf{z}\}$. Therefore, skills do not depend on a single time step, but are rather defined at trajectory-level. Although closely related, we can show that the feature space $\Phi$ is a subset of the skill space $\mathcal{Z}$, which forms the convex hull of the feature space. Therefore, the skill space is larger than the goal space, $\Phi \subset \mathcal{Z} = \text{Conv}(\Phi)$.

## 4 METHODS

In this section, we introduce a Skill-Conditioned Optimal Agent (SCOPA), an approximate solution to Problem P1. In the following section, we define the objective functions SCOPA considers for optimization. Then, we describe the design choices made to efficiently optimize the objectives.

### 4.1 ACTOR OPTIMIZATION

First, we relax the constraint from Problem P1 using the $L^2$ norm and $\delta$, a threshold that quantifies the maximum acceptable distance between the desired skill and the expected features,

$$\forall t, \forall \mathbf{z} \in \mathcal{Z}, \quad \text{maximize } \mathbb{E}_{\pi(\cdot|\cdot, \mathbf{z})} \left[ \sum_{i=0}^{\infty} \gamma^i r_{t+i} \right] \quad \text{subject to } \left\| \mathbb{E}_{\pi(\cdot|\cdot, \mathbf{z})} [\boldsymbol{\phi}(s)] - \mathbf{z} \right\|_2 \leq \delta \quad \text{(P2)}$$

---

**Algorithm 1** SCOPA

---

1: Initialize dataset $\mathcal{D}$ with samples collected from random exploration, and set up parameters $\theta$
2: **while** not converged **do**
3:     Sample $\mathbf{z} \sim \mathcal{U}(\mathcal{Z})$
4:     **for** $T$ steps **do**
        ▷ *Environment exploration steps*
5:         Sample action $a_t \sim \pi(\cdot|\widetilde{s}_t, \mathbf{z})$            ▷ *$\widetilde{s}_t$ denotes the hidden state of the world model $\mathcal{W}$*
6:         $r_t, s_{t+1} \leftarrow$ perform action $a_t$ in environment
7:         Add transition $(s_t, a_t, r_t, \boldsymbol{\phi}_t, s_{t+1})$ to its corresponding sequence in dataset $\mathcal{D}$     ▷ $\boldsymbol{\phi}_t := \phi(s_t)$
        ▷ *Training steps*
8:         Train world model parameters $\theta_{\mathcal{W}}$ to reconstruct observations, rewards $\widehat{r}$ and features $\widehat{\boldsymbol{\phi}}$ using $\mathcal{D}$
9:         Sample $N$ skills $\widetilde{\mathbf{z}}_i \sim \mathcal{U}(\mathcal{Z})$ and states $s_i \in \mathcal{D}$; compute latent states $\widetilde{s}_i \leftarrow \mathcal{W}(s_i)$
10:        For each $i$, perform a rollout of horizon $H$ in imagination using $\mathcal{W}$ starting from $\widetilde{s}_i$ with $\pi(\cdot|\cdot, \widetilde{\mathbf{z}}_i)$
11:        $\theta_\lambda \leftarrow \theta_\lambda - \alpha_\lambda \nabla L(\theta_\lambda)$            ▷ *Minimize Lagrange multiplier loss (3)*
12:        $\theta_\pi \leftarrow \theta_\pi + \alpha_\pi \nabla J(\theta_\pi)$               ▷ *Maximize actor's objective (4)*
13:        Update parameters $\theta_v$ and $\theta_\psi$ of value and successor features networks

---

Second, we derive an upper bound for the distance between the desired skill and the expected features, whose proof is provided in Appendix B. A similar proposition is proven in a more general case in Appendix B.1. The goal is to minimize the bound so that the constraint in Problem P2 is satisfied.

**Proposition.** *Consider an infinite horizon, finite MDP with observable features in $\Phi$. Let $\pi$ be a policy and let $\psi$ be the discounted successor features. Then, for all skills $\mathbf{z} \in \mathcal{Z}$, we can derive an upper bound for the distance between $\mathbf{z}$ and the expected features under $\pi$:*

$$\left\| \mathbb{E}_{\pi(\cdot|\cdot, \mathbf{z})}\left[\phi(s)\right] - \mathbf{z} \right\|_2 \leq \mathbb{E}_{\pi(\cdot|\cdot, \mathbf{z})}\left[\left\|(1-\gamma)\boldsymbol{\psi}(s, \mathbf{z}) - \mathbf{z}\right\|_2\right] \tag{1}$$

Third, we derive a new Problem P3 by replacing the intractable constraint of Problem P2 with the tractable upper bound in Equation 1. Note that if the more restrictive constraint in Problem P3 is satisfied, then the constraint in Problem P2 is necessarily satisfied as well.

$$\forall \mathbf{z} \in \mathcal{Z}, \quad \text{maximize } \mathbb{E}_{\pi(\cdot|\cdot, \mathbf{z})}\left[v(s, \mathbf{z})\right] \quad \text{subject to } \mathbb{E}_{\pi(\cdot|\cdot, \mathbf{z})}\left[\left\|(1-\gamma)\boldsymbol{\psi}(s, \mathbf{z}) - \mathbf{z}\right\|_2\right] \leq \delta \tag{P3}$$

Finally, we solve Problem P3 by combining the objective and the constraint using Lagrangian relaxation as described by Abdolmaleki et al. (2018; 2023). For all states $s$, and all skills $\mathbf{z} \in \mathcal{Z}$, we maximize the following objective:

$$(1 - \lambda_{\mathcal{L}}(s, \mathbf{z}))\, v(s, \mathbf{z}) - \lambda_{\mathcal{L}}(s, \mathbf{z})\|(1-\gamma)\boldsymbol{\psi}(s, \mathbf{z}) - \mathbf{z}\|_2 \quad \text{subject to } 0 \leq \lambda_{\mathcal{L}}(s, \mathbf{z}) \leq 1 \tag{2}$$

The first term in red aims at maximizing the return, while the second term in blue aims at executing the desired skill. We extend generalized policy iteration to successor features by alternating policy evaluation for both critics $v(s, \mathbf{z})$ and $\boldsymbol{\psi}(s, \mathbf{z})$, and policy improvement based on Equation 2. Thus, we introduce a policy (skill) improvement with successor features that is analogous to the classic policy (performance) improvement with value function.

The coefficient $\lambda_{\mathcal{L}}$ is analogous to a Lagrange multiplier and is optimized to balance the quality-diversity trade-off. If $\|(1-\gamma)\boldsymbol{\psi}(s_1, \mathbf{z}_1) - \mathbf{z}_1\|_2 \leq \delta$ is satisfied for $(s_1, \mathbf{z}_1)$, we expect $\lambda_{\mathcal{L}}(s_1, \mathbf{z}_1)$ to decrease to encourage maximizing the return. On the contrary, if the constraint is not satisfied for $(s_2, \mathbf{z}_2)$, we expect $\lambda_{\mathcal{L}}(s_2, \mathbf{z}_2)$ to increase to encourage satisfying the constraint.

We represent the function $\lambda_{\mathcal{L}}$ by a neural network parameterized by $\theta_\lambda$. We optimize the parameters $\theta_\lambda$ so that $\lambda_{\mathcal{L}}(s, \mathbf{z}) \to 1$ when the actor is unable to solve the constraint, in order to focus on satisfying the constraint. Conversely, $\lambda_{\mathcal{L}}(s, \mathbf{z}) \to 0$ when the constraint is satisfied, to focus primarily on maximizing the return. In practice, we use a cross-entropy loss to optimize those parameters:

$$L(\theta_\lambda) = \mathbb{E}_{\substack{s \sim \rho \\ \mathbf{z} \sim \mathcal{U}(\mathcal{Z})}}\left[-(1-y)\log(1 - \lambda_{\mathcal{L}}(s, \mathbf{z})) - y\log(\lambda_{\mathcal{L}}(s, \mathbf{z}))\right]$$

$$\text{where } y = \begin{cases} 0 & \text{if} \quad \|(1-\gamma)\boldsymbol{\psi}(s, \mathbf{z}) - \mathbf{z}\|_2 \leq \delta \\ 1 & \text{otherwise} \end{cases} \tag{3}$$

## 4.2 TRAINING

The main challenge to train skill-conditioned function approximators like our policy or the value function/successor features is the need to sample a large number of $(s, \mathbf{z})$ combinations (Schaul

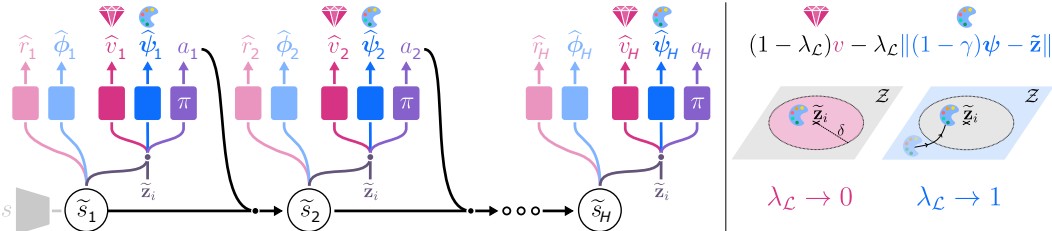

Figure 1: **Left** Imagination rollout performed within the world model $\mathcal{W}$. Each individual rollout $i$ generates on-policy transitions following skill $\widetilde{\mathbf{z}}_i$, starting from a state $\widetilde{s}_1$ for a fixed number of steps $H$. The world model predicts $\widehat{r}_i$ and $\widehat{\phi}_i$ that enable to compute $\widehat{v}_i$ and $\widehat{\psi}_i$ respectively. **Right** Actor's objective function computed with the predicted $\widehat{v}$ and $\widehat{\psi}$. The Lagrangian multiplier is optimized to balance the quality-diversity trade-off, see Equation 3.

et al., 2015; Borsa et al., 2018), see Section 2.2. To solve this problem, we use the model-based RL algorithm DREAMERV3 (Hafner et al., 2023) that enables to generate thousands of on-policy trajectories simultaneously in imagination.

A pseudo-code of SCOPA is provided in Algorithm 1. At each iteration, a skill $\mathbf{z}$ is uniformly sampled and for $T$ steps, the agent interacts with the environment following skill $\mathbf{z}$ with $\pi(\cdot|\cdot, \mathbf{z})$. At each step, the transition is stored in a dataset $\mathcal{D}$, which is used to perform a world model training step. Then, $N$ skills are uniformly sampled to perform rollouts in imagination, and those rollouts are used to (1) train the two critics $v(s, \mathbf{z})$, $\psi(s, \mathbf{z})$ and (2) train the actor $\pi$.

**World model training** The dataset is used to train the world model $\mathcal{W}$ according to DreamerV3. In addition to the reward $\widehat{r}_t$, we extend the model to estimate the features $\widehat{\phi}_t$, like shown on Figure 1.

**Critic training** The estimated rewards $\widehat{r}_t$ and features $\widehat{\phi}_t$ predicted by the world model are used to predict the value function $\widehat{v}$ and the successor features $\widehat{\psi}$ respectively. Then, similarly to DREAMERV3, the value function $\widehat{v}$ and successor features $\widehat{\psi}$ are trained to regress the $\lambda$-returns, $V_\lambda$ and $\Psi_\lambda$ respectively. The successor features target is defined recursively as follows:

$$\Psi_\lambda(\widetilde{s}_t, \widetilde{\mathbf{z}}) = \widehat{\phi}_t + \gamma \widehat{c}_t \left( (1 - \lambda) \widehat{\psi}(\widetilde{s}_{t+1}, \widetilde{\mathbf{z}}) + \lambda \Psi_\lambda(\widetilde{s}_{t+1}, \widetilde{\mathbf{z}}) \right) \quad \text{and} \quad \Psi_\lambda(\widetilde{s}_H, \widetilde{\mathbf{z}}) = \widehat{\psi}(\widetilde{s}_H, \widetilde{\mathbf{z}})$$

**Actor training** For each actor training step, we sample $N$ skills $\widetilde{\mathbf{z}}_1 \ldots \widetilde{\mathbf{z}}_N \in \mathcal{Z}$. We then perform $N$ rollouts of horizon $H$ in imagination using the world model and policies $\pi(\cdot|\cdot, \widetilde{\mathbf{z}}_i)$. Those rollouts are used to train the critic $v$, the successor features network $\psi$, and the actor by backpropagating through the dynamics of the model. The actor maximizes the following objective, with an entropy regularization for exploration, where sg $(\cdot)$ represents the *stop gradient* function.

$$J(\theta_\pi) = \mathbb{E}_{\substack{\widetilde{s}_{1:H} \sim \mathcal{W}, \pi \\ \widetilde{\mathbf{z}} \sim \mathcal{U}(\mathcal{Z})}} \left[ \sum_{t=1}^{H} (1 - \text{sg}(\lambda_{\mathcal{L}, t})) \underbrace{V_\lambda(\widetilde{s}_t, \widetilde{\mathbf{z}})}_{\text{Performance}} - \text{sg}(\lambda_{\mathcal{L}, t}) \underbrace{\|(1 - \gamma)\Psi_\lambda(\widetilde{s}_t, \widetilde{\mathbf{z}}) - \widetilde{\mathbf{z}}\|_2}_{\text{Distance to desired skill } \widetilde{\mathbf{z}}} \right] \quad (4)$$

## 5 EXPERIMENTS

### 5.1 TASKS

We compare our method to prior techniques on a range of challenging continuous control tasks using Google Brax (Freeman et al., 2021) physics engine. We consider three classic locomotion environments called Walker, Ant and Humanoid that we combine with four different feature functions that we call *feet contact*, *jump*, *velocity* and *angle*. In these traditional locomotion tasks, the objective is to go forward as fast as possible while minimizing energy consumption.

The feet contact is a vector indicating for each foot of the agent, if the foot is in contact or not with the ground, exactly as defined by Cully et al. (2015) and in DCG-ME (Faldor et al., 2023). For example,

if the Ant only touches the ground with its second foot at time step $t$, then $\phi(s_t) = \begin{bmatrix} 0 & 1 & 0 & 0 \end{bmatrix}^\mathsf{T}$. The feet contact have been extensively studied in Quality-Diversity (Cully et al., 2015; Nilsson & Cully, 2021; Faldor et al., 2023) to learn diverse locomotion behaviors with different gaits. The diversity of feet contact found by such QD algorithms has been demonstrated to be very useful in downstream tasks such as damage recovery (Cully et al., 2015). Note that to execute a specific skill, the agent needs multiple time steps, see Section 3 for a longer discussion.

The jump feature is a one-dimensional vector corresponding to the height of the lowest foot. For example, if the left foot of the humanoid is 10 cm above the ground and if its right foot is 3.5 cm above the ground, then the features $\phi(s_t) = [0.035]$. The skills derived from the jump features are also challenging to execute because to maintain an average $\mathbf{z} = \frac{1}{T} \sum_{i=0}^{T-1} \phi_{t+i}$, the agent is forced to oscillate around that value $\mathbf{z}$ because of gravity.

The velocity feature is a two-dimensional vector giving the velocity of the agent in the $xy$-plane, $\phi(s_t) = \begin{bmatrix} v_x(t) & v_y(t) \end{bmatrix}^\mathsf{T}$. The velocity feature space has been extensively studied in GCRL (Liu et al., 2022; Zhu et al., 2021; Finn et al., 2017). We evaluate on the velocity features to show that our method works on classic GCRL tasks. The velocity features are also interesting because satisfying a velocity goal that is negative on the $x$-axis is directly opposite to maximizing the return that is defined as forward velocity minus energy consumption.

Finally, the angle is a two-dimensional vector representing the angle $\alpha$ of the main body about the $z$-axis, $\phi(s_t) = \begin{bmatrix} \cos(\alpha) & \sin(\alpha) \end{bmatrix}^\mathsf{T}$. You can find the feature spaces summarized in Appendix C.1.

## 5.2 BASELINES

We compare SCOPA against four baselines SMERL (Kumar et al., 2020), REVERSE SMERL (Zahavy et al., 2022), DCG-ME (Faldor et al., 2023) and UVFA (Schaul et al., 2015) that all return a skill-conditioned policy balancing a quality-diversity trade-off.

SMERL learns a repertoire of policies that maximize the expected return while also producing distinct behaviors by encouraging the trajectories of different latent variables $\mathbf{z}$ to be distinct. To that end, it uses the following reward: $r_{\text{SMERL}} = r + \alpha \mathbb{1}(R \geq R^* - \epsilon)\tilde{r}$. Similarly, REVERSE SMERL focuses on finding diverse behaviours, and only optimizes for performance if the policy is not close to the optimal policy: $r_{\text{REVERSE}} = \mathbb{1}(R < R^* - \epsilon)r + \alpha\tilde{r}$. For a fair comparison with our method, we compute the diversity reward with continuous DIAYN (Choi et al., 2021) so that SMERL learns a continuous (infinite) subspace instead of a discrete (finite) subset of policies (Kumar et al., 2020). Finally, we use DIAYN+prior (Eysenbach et al., 2018; Chalumeau et al., 2022) with $q(\mathbf{z}_{\text{DIAYN}}|s_t)$ and feature priors to guide SMERL and REVERSE SMERL towards relevant skills as explained in DIAYN original paper. More details about the baselines are available in Appendix D.1. DCG-ME is a state-of-the-art Quality-Diversity algorithm for challenging continuous control tasks that evolves a population of both high-performing and diverse solutions, and simultaneously distills those solutions into a single skill-conditioned policy. We also evaluate the performance of SCOPA against a standard GCRL algorithm, which rely on Universal Value Function Approximators (UVFA). UVFA is trained similar to SCOPA but the Lagrangian multiplier $\lambda_{\mathcal{L}}$ is fixed, and maximizes a reward that balances between task reward and following the skill at each time step: $(1 - \lambda_{\mathcal{L}})r_t - \lambda_{\mathcal{L}} \|\phi_t - \mathbf{z}\|_2$.

We perform two additional ablation experiments, that we call Separable-$\mathbf{z}$ (SEP-$\mathbf{z}$) and FIXED-$\lambda$. For SEP-$\mathbf{z}$, we remove the successor features representation and use a naive distance to skill instead $\sum_t \gamma^t \|\phi_t - \mathbf{z}\|_2$. For FIXED-$\lambda$, we remove the Lagrangian multiplier, so the value of $\lambda_{\mathcal{L}}$ remains constant. Thus, SEP-$\mathbf{z}$ and FIXED-$\lambda$ bridge the gap between SCOPA and UVFA. A summarized description of all variants under study is provided in Table D.3.

## 5.3 EVALUATION METRICS

We evaluate our method using two types of metrics: (1) the distance to skill metrics that evaluate the ability of an agent to execute target skills, and (2) the performance metrics that quantify the ability of an agent to maximize return while executing target skills. Each experiment is replicated 10 times with random seeds independently and trains all components from scratch. We report the Inter-Quartile Mean (IQM) value for each metric, with the estimated 95% Confidence Interval (CI) (Agarwal et al., 2021). The statistical significance of the results is evaluated using the Mann-Whitney $U$ test (Mann & Whitney, 1947) and the probabilities of improvement are reported in Appendix A.1.

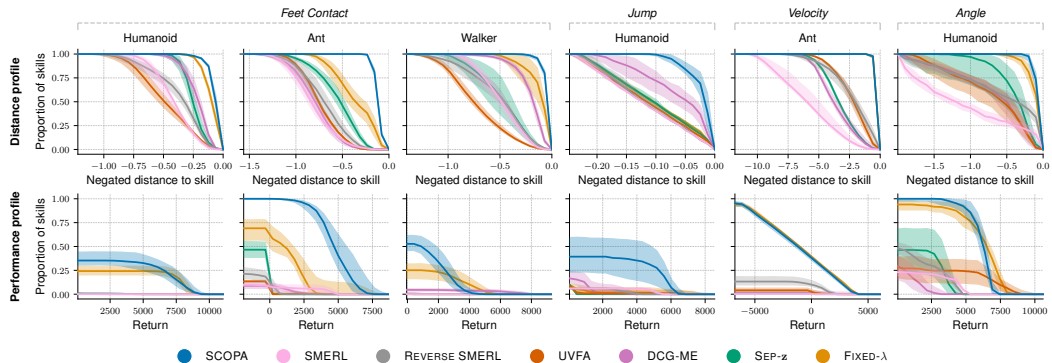

Figure 2: (**top**) Distance profiles and (**bottom**) performance profiles for each task defined in Section 5.3. The lines represent the IQM for 10 replications, and the shaded areas correspond to the 95% CI. Figure F.22 illustrates how to read distance and performance profiles.

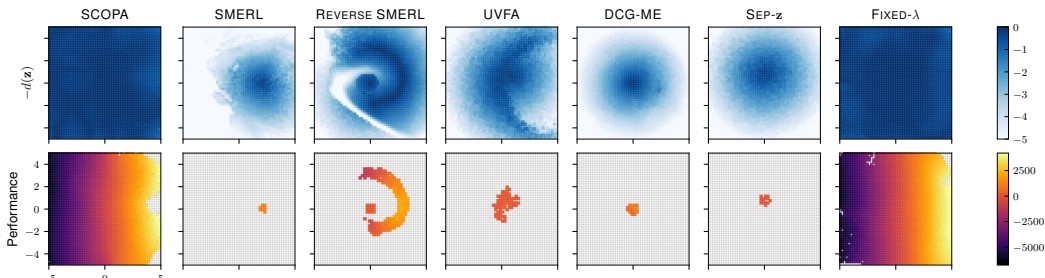

Figure 3: Heatmaps of (**top**) negative distance to skill, (**bottom**) performance defined in Section 5.3, for Ant Velocity. The heatmap represents the skill space $\mathcal{Z} = [-5 \text{ m/s}, 5 \text{ m/s}]^2$, of target velocities. This space is discretized into cells, with each cell representing a distinct skill $\mathbf{z} = [v_x \quad v_y]^\mathsf{T}$. In the bottom row, empty cells show which skills are not successfully executed (i.e. $d(\mathbf{z}) > d_{\text{eval}}$).

**Distance to skill metrics** To evaluate the ability of a policy to achieve a given skill $\mathbf{z}$, we estimate the *expected distance to skill*, denoted $d(\mathbf{z})$, by averaging the euclidean distance between the desired skill $\mathbf{z}$ and the observed skill over 10 rollouts. First, we use $d(\mathbf{z})$ to compute *distance profiles* on Figure 2, which quantify for a given distance $d$, the proportion of skills in the skill space that have an expected distance to skill smaller than $d$, computed with the function $d \mapsto \frac{1}{N_\mathbf{z}} \sum_{i=1}^{N_\mathbf{z}} \mathbb{1}(d(\mathbf{z}_i) < d)$. Second, we summarize the ability of a policy to execute skills with the *distance score*, $\frac{1}{N_\mathbf{z}} \sum_{i=1}^{N_\mathbf{z}} -d(\mathbf{z}_i)$.

**Performance metrics** To evaluate the ability of a policy to solve a task given a skill $\mathbf{z}$, we estimate the *expected undiscounted return*, denoted $R(\mathbf{z})$, by averaging the return over 10 rollouts. First, we use $R(\mathbf{z})$ to compute *performance profiles* on Figure 2, which quantify for a given return $R$, the proportion of skills in the skill space that have an expected return larger than $R$, after filtering out the skills that are not achieved by the policy. To this end, we compute the expected distance to skill $d(\mathbf{z})$, and discard skills with an expected distance to skill that is larger than a predefined threshold, $d(\mathbf{z}) > d_{\text{eval}}$. More precisely, the performance profile is the function $R \mapsto \frac{1}{N_\mathbf{z}} \sum_{i=1}^{N_\mathbf{z}} \mathbb{1}(d(\mathbf{z}_i) < d_{\text{eval}}, R(\mathbf{z}_i) > R)$. Second, we summarize the ability of a policy to maximize return while executing skills, with the *performance score*, $\frac{1}{N_\mathbf{z}} \sum_{i=1}^{N_\mathbf{z}} R(\mathbf{z}) \mathbb{1}(d(\mathbf{z}_i) < d_{\text{eval}})$.

## 5.4 RESULTS

The goal of our experiments is to answer two questions: (1) Is SCOPA capable of solving problem P1? (2) Are the high-performing and diverse skills useful to adapt to new MDPs? Our results show that SCOPA achieves significantly better distance and performance scores compared to other baselines (Fig. 4). Empirical evaluations also show that SCOPA can adapt to out-of-distribution changes, in a few shots or with hierarchical learning. Videos of our results are available at: bit.ly/scopa.

**Distance to skill**   SCOPA outperforms all baselines in executing a wide range of skills (Fig. 2). SMERL, REVERSE SMERL and UVFA minimize the distance between features and skills. However, for challenging skill spaces like feet contact, this approach cannot achieve most of the skills, see Section 3. This is why they can only execute skills in the corners of the skill space where $\mathbf{z} = \phi_t$, as shown in Figure A.9. Notably, SCOPA is capable of achieving skills that are contrary to the task reward, as illustrated by the velocity features in Figure 2 and Figure 3, which is not the case for SMERL or REVERSE SMERL. SCOPA outperforms SMERL in all tasks since SMERL primarily optimizes for the task reward rather than optimizing to acquire skills. Finally, our approach outperforms DCG-ME that fails to explicitly minimize the expected skill distance, a common issue among QD algorithms (Flageat & Cully, 2023). These comparisons highlight the significance of the policy skill improvement term in blue in Equation 4.

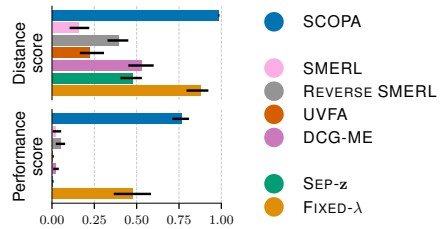

Figure 4: Distance and performance scores normalized and aggregated across all tasks. The values correspond to the IQM while the error bars represent IQM 95% CI.

**Performance**   SCOPA outperforms most baselines in maximizing return (Fig. 4), as they don't achieve many skills in the first place, and the performance score only evaluates the performance of skills successfully executed by the policy. FIXED-$\lambda$ is a variation of our method that does not adapt the Lagrange multiplier and is the only baseline that gets performance scores and profiles comparable to SCOPA. However, SCOPA is capable to learn to hop on one foot in Humanoid feet contact. In contrast, FIXED-$\lambda$ covers a smaller range of skills, as evidenced by the edges of the skill space on Figure A.9. Additionally, SCOPA significantly outperforms FIXED-$\lambda$ on the challenging jump task, as demonstrated in Figure 2, due to the necessity of a strong weight on the constraint. Ultimately, using an adaptive $\lambda_{\mathcal{L}}$ proves advantageous for our approach.

**Using skills for adaptation**   We evaluate on five downstream tasks, namely Humanoid - Hurdles, Humanoid - Motor Failure, Ant - Gravity, Walker - Friction and Ant - Obstacle. On Ant - Obstacle, we hierarchically learn a meta-controller to choose the skills of a pre-trained policy. On all tasks except Ant - Obstacle, we perform few-shot adaptation experiments where no re-training is allowed and where we evaluate the top-performing skills for each method, see Appendix C.2 for more details. SCOPA is consistently as robust or more robust than other baselines on all downstream tasks except on Walker - Friction where DCG-ME is better, see Figure 5. On the hurdles task, SCOPA significantly outperforms other baselines by consistently jumping over higher hurdles. On the motor failure task, SCOPA and UVFA show great robustness on a wide range of damage strengths but perform worse than SMERL and REVERSE SMERL for maximal damage. Finally, on the obstacle task, SCOPA significantly outperforms other baselines.

## 6   RELATED WORK

**Learning Diverse Optimal Policies**   Quality-Diversity (QD) optimization is a family of evolutionary algorithms that generate large collections of solutions, such as policies, that are both diverse and high-performing (Pugh et al., 2016; Cully & Demiris, 2018). A feature space (also called descriptor space) is defined to evaluate the diversity of the solutions (Cully et al., 2015). QD algorithms have been combined with RL-based methods (Pierrot et al., 2022; Nilsson & Cully, 2021; Faldor et al., 2023; Tjanaka et al., 2022; Fontaine et al., 2020) and have been shown to be effective for skill discovery (Chalumeau et al., 2022). QD approaches and our framework both solve the same problem, however most QD algorithms output a large number of policies instead of a single skill-conditioned one, which can be difficult to deal with in practice in downstream tasks (Faldor et al., 2023). Finally, QD algorithms rely on Genetics Algorithms which can cause slow convergence in challenging environments with high-dimensional state spaces (Pierrot et al., 2022). Similarly to QD methods, DOMINO (Zahavy et al., 2022) learns a set of diverse and near-optimal policies, but contrary to SCOPA, the policies discovered by DOMINO are not trained to execute specific target skills $\mathbf{z}$.

Figure 5: Performance for each algorithm in environments with different levels of perturbations after few-shot adaptation or after hierarchical learning. The lines represent the IQM for 10 replications, and the shaded areas correspond to the 95% CI.

**Unsupervised Reinforcement Learning** Most URL approaches discover diverse skills by maximizing an intrinsic reward that is defined in terms of the discriminability of the trajectories induced by the different skills. Usually, the methods maximize the Mutual Information (MI) between the trajectories and the skills $I(\tau, \mathbf{z})$ (Gregor et al., 2016; Sharma et al., 2019; Mazzaglia et al., 2022), which is usually reduced to a MI-maximization between skills and states with the following lower bound: $I(\tau, \mathbf{z}) \geq \sum_{t=1}^{T} I(s_t, \mathbf{z})$. Moreover, those methods may rely on the use of priors to focus the exploration on useful skills (Eysenbach et al., 2018; Sharma et al., 2019). In addition to priors, some approaches like SMERL (Kumar et al., 2020) or DIAYN+REWARD and DADS+REWARD (Chalumeau et al., 2022) maximize a trade-off between the extrinsic reward from the task, and an intrinsic reward for diversity, further reducing the gap between those methods and our algorithm. Other approaches, such as (Park et al., 2022; 2023), manage to discover diverse dynamic skills without any prior on $\phi$; nonetheless, these approaches do not maximize any extrinsic reward.

**Goal-Conditioned Reinforcement Learning** GCRL trains agents to achieve different goals, which are commonly defined as a subspace of the states, $\mathbf{g} = \phi(s)$. However, some approaches don't ignore the reward of the original problem to learn useful policies (Sutton et al., 2023). Additionally, GCRL can be reformulated in a maximum mutual information setting, further reducing the gap between GCRL and skill discovery (Choi et al., 2021; Gu et al., 2021). In our method, if we ignore the extrinsic reward and optimize only to satisfy the constraint, we can show that our problem is related to GCRL. Moreover, in Goal-Augmented MDP (GA-MDP), the reward $r(s_t, a_t, \mathbf{g}) = \mathbb{1}(\left\|\phi_{t+1} - \mathbf{g}\right\|_2 \leq \delta)$ is sparse (Andrychowicz et al., 2017) and reshaped dense rewards like $r(s_t, a_t, \mathbf{g}) = -d(\phi_{t+1}, \mathbf{g})$ cause local optima challenges (Trott et al., 2019), which justifies the successor features policy skill improvement introduced in Section 4. Another line of work learns diverse exploring options, which can then be used in a temporally-extended manner to improve the efficiency of RL algorithms (Klissarov & Machado, 2023).

## 7 CONCLUSION

In this work, we present a quality-diversity reinforcement learning algorithm, SCOPA, that learns to execute a continuous range of skills while maximizing return. We extend the generalized policy iteration framework with a policy skill improvement update that is analogous to the classic policy improvement update. Our approach seamlessly unifies value function and successor features policy iteration with constrained optimization to (1) maximize performance, while (2) executing skills.

Our empirical evaluations suggest that SCOPA outperforms several skill-conditioned reinforcement learning and quality-diversity methods on a variety of continuous control locomotion tasks. Quantitative results demonstrate that SCOPA can extrapolate to out-of-distribution test conditions, while qualitative analyses showcase a range of impressive behaviors.

In this paper, the feature functions are defined in advance, and provide the user the opportunity to give the algorithm prior information about potential downstream tasks, as well as more control and interpretability for the skills. However, an exciting direction for future work would be to learn the feature functions in an unsupervised manner, broadening the scope to include task-agnostic skills.

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

# A SUPPLEMENTARY RESULTS

## A.1 QUANTITATIVE RESULTS

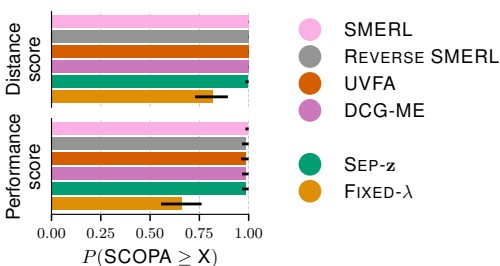

Figure A.6: Probabilities of improvement of SCOPA over all other baselines, aggregated across all tasks, as defined by Agarwal et al. (2021).

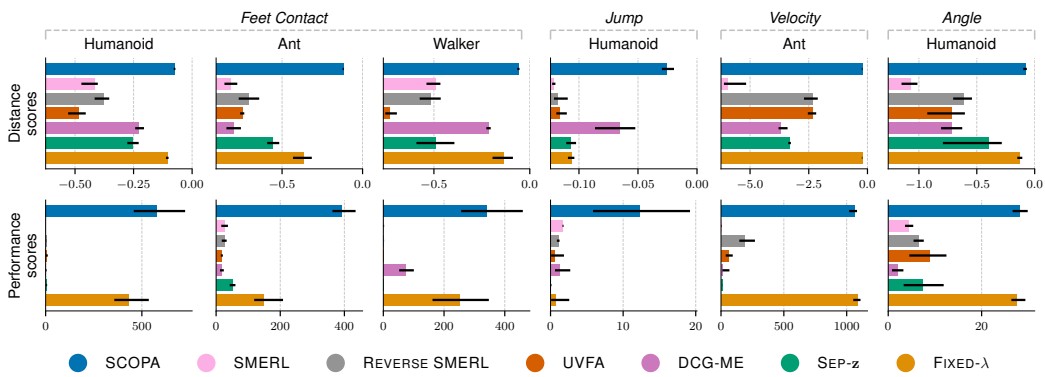

Figure A.7: IQM for distance and performance scores per task.

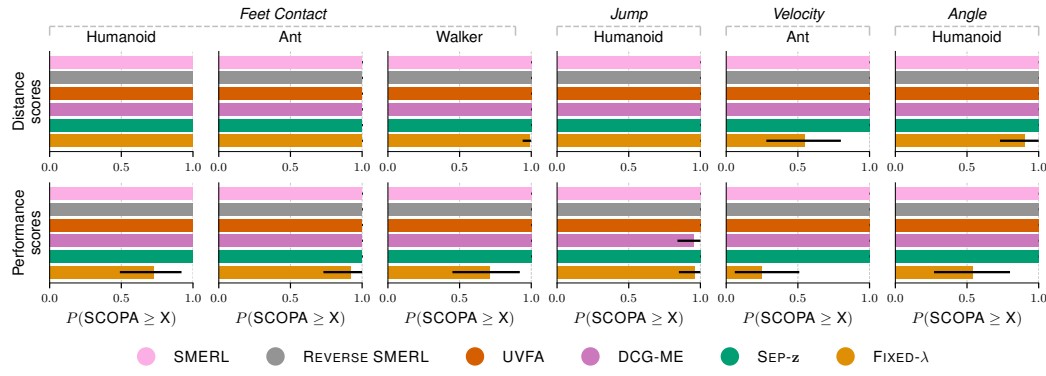

Figure A.8: Per-task probabilities of improvement (as defined by Agarwal et al. (2021)) of SCOPA over all other baselines.

## A.2 HEATMAPS

In Figures A.9 to A.14, we report the profiles and heatmaps defined in Section 5.3. In the bottom row, empty cells show which skills are not successfully executed (i.e. $d(\mathbf{z}) > d_{\text{eval}}$).

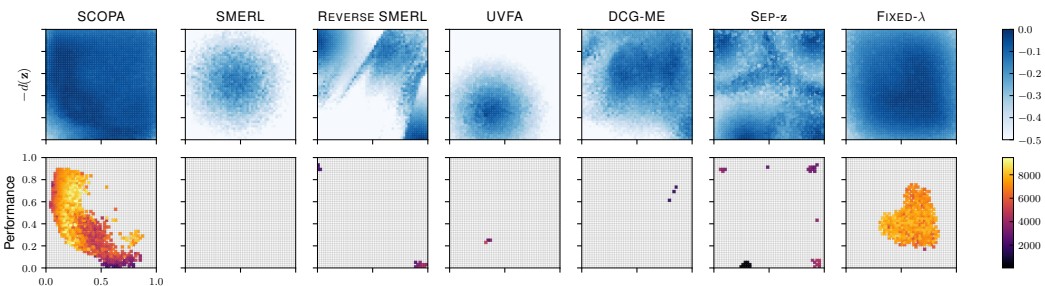

Figure A.9: **Humanoid Feet Contact** Heatmaps of **(top)** negative distance to skill, **(bottom)** performance defined in Section 5.3 for all baselines. In the bottom row, empty cells show which skills are not successfully executed (i.e. $d(\mathbf{z}) > d_{\text{eval}}$).

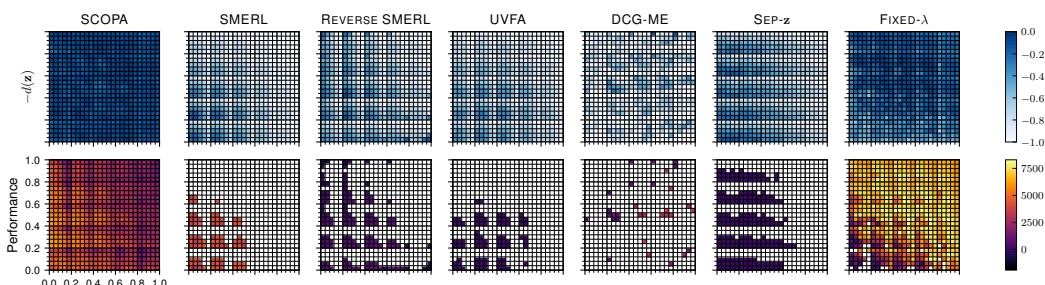

Figure A.10: **Ant Feet Contact** Heatmaps of **(top)** negative distance to skill, **(bottom)** performance defined in Section 5.3 for all baselines. In the bottom row, empty cells show which skills are not successfully executed (i.e. $d(\mathbf{z}) > d_{\text{eval}}$).

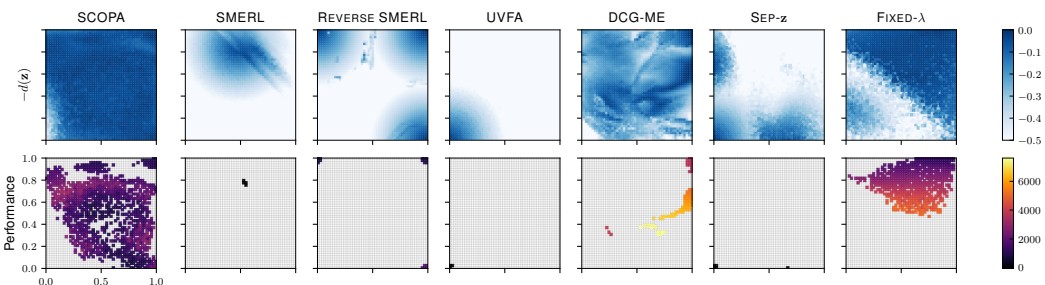

Figure A.11: **Walker Feet Contact** Heatmaps of **(top)** negative distance to skill, **(bottom)** performance defined in Section 5.3 for all baselines. In the bottom row, empty cells show which skills are not successfully executed (i.e. $d(\mathbf{z}) > d_{\text{eval}}$).

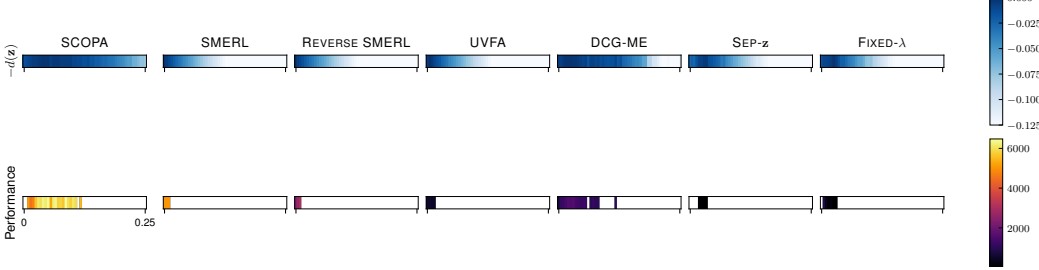

Figure A.12: **Humanoid Jump** Heatmaps of **(top)** negative distance to skill, **(bottom)** performance defined in Section 5.3 for all baselines. In the bottom row, empty cells show which skills are not successfully executed (i.e. $d(\mathbf{z}) > d_{\text{eval}}$).

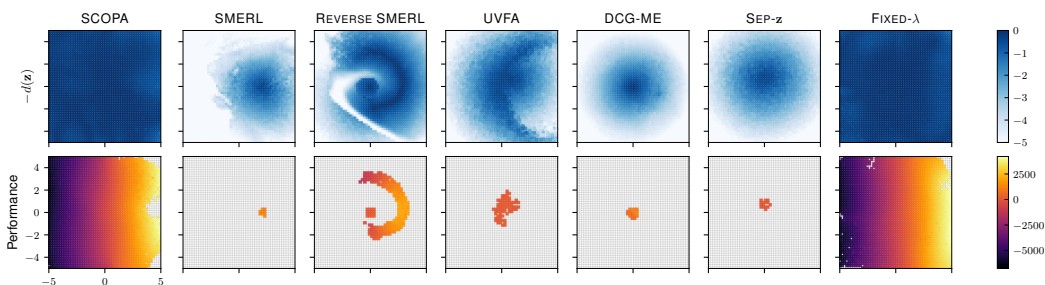

Figure A.13: **Ant Velocity** Heatmaps of **(top)** negative distance to skill, **(bottom)** performance defined in Section 5.3 for all baselines. In the bottom row, empty cells show which skills are not successfully executed (i.e. $d(\mathbf{z}) > d_{\text{eval}}$).

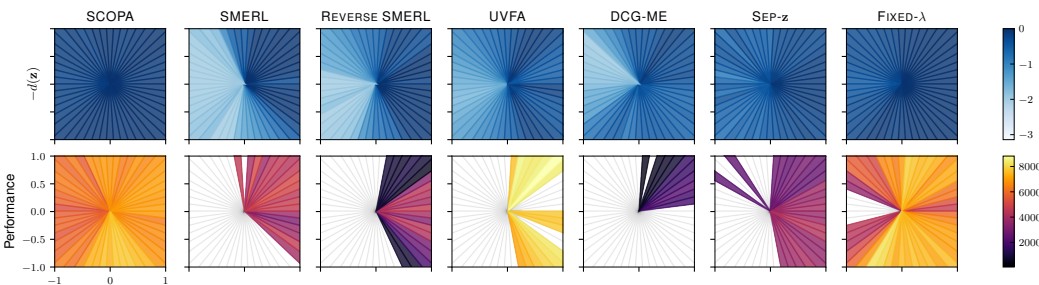

Figure A.14: **Humanoid Angle** Heatmaps of **(top)** negative distance to skill, **(bottom)** performance defined in Section 5.3 for all baselines. In the bottom row, empty cells show which skills are not successfully executed (i.e. $d(\mathbf{z}) > d_{\text{eval}}$).

## A.3 RESULTS WITHOUT FILTERING WITH $d_{\text{EVAL}}$

In Figures A.15 to A.21, we report the profiles and heatmaps defined in Section 5.3 except that skills that are not successfully executed (i.e. $d(\mathbf{z}) > d_{\text{eval}}$) are **not** filtered out.

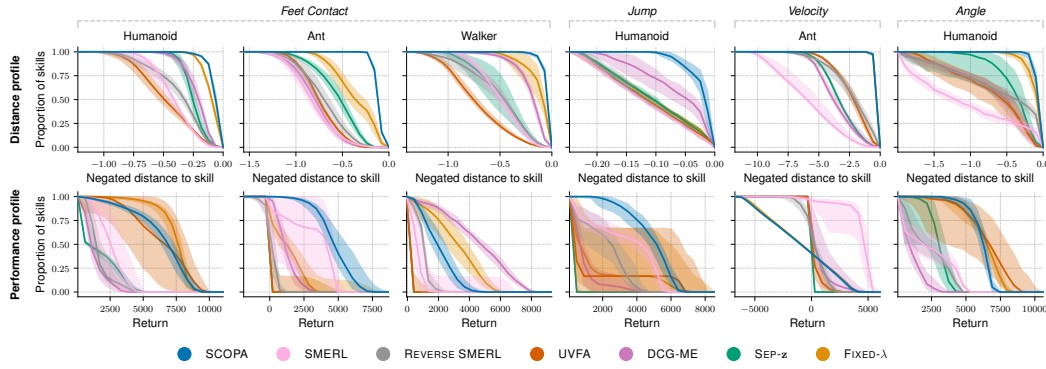

Figure A.15: (**top**) Distance profiles and (**bottom**) performance profiles for each task defined in Section 5.3 similar to Figure 2 except that skills that are not successfully executed (i.e. $d(\mathbf{z}) > d_{\text{eval}}$) are **not** filtered out. The lines represent the IQM for 10 replications, and the shaded areas correspond to the 95% CI. Figure F.22 illustrates how to read distance and performance profiles.

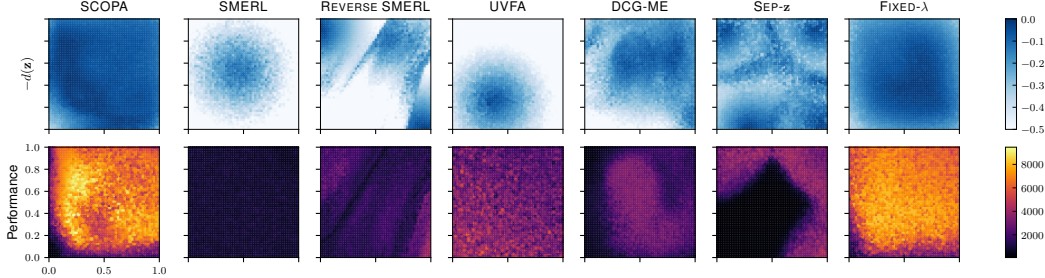

Figure A.16: **Humanoid Feet Contact** Heatmaps of (**top**) negative distance to skill, (**bottom**) performance defined in Section 5.3 for all baselines. In the bottom row, the skills that are not successfully executed (i.e. $d(\mathbf{z}) > d_{\text{eval}}$) are **not** filtered out.

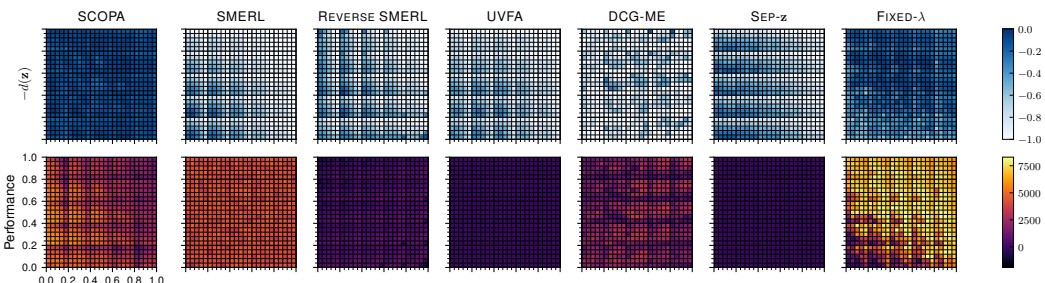

Figure A.17: **Ant Feet Contact** Heatmaps of (**top**) negative distance to skill, (**bottom**) performance defined in Section 5.3 for all baselines. In the bottom row, the skills that are not successfully executed (i.e. $d(\mathbf{z}) > d_{\text{eval}}$) are **not** filtered out.

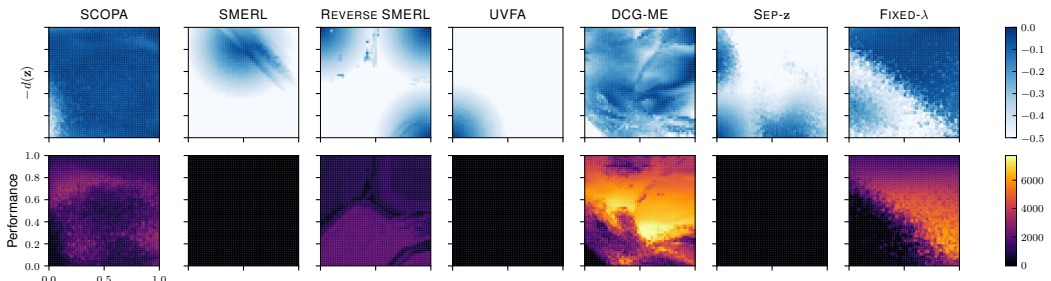

Figure A.18: **Walker Feet Contact** Heatmaps of **(top)** negative distance to skill, **(bottom)** performance defined in Section 5.3 for all baselines. In the bottom row, the skills that are not successfully executed (i.e. $d(\mathbf{z}) > d_{\text{eval}}$) are **not** filtered out.

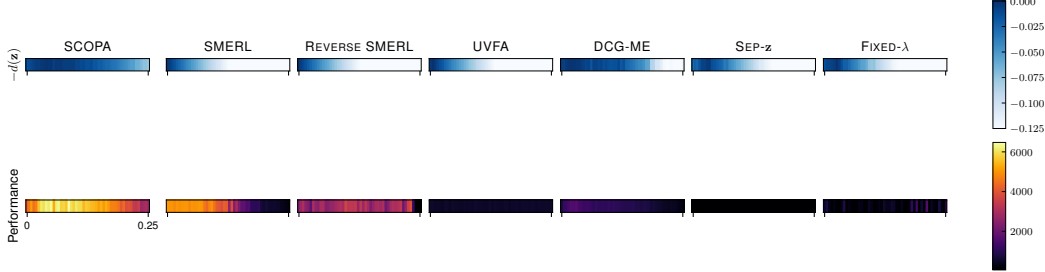

Figure A.19: **Humanoid Jump** Heatmaps of **(top)** negative distance to skill, **(bottom)** performance defined in Section 5.3 for all baselines. In the bottom row, the skills that are not successfully executed (i.e. $d(\mathbf{z}) > d_{\text{eval}}$) are **not** filtered out.

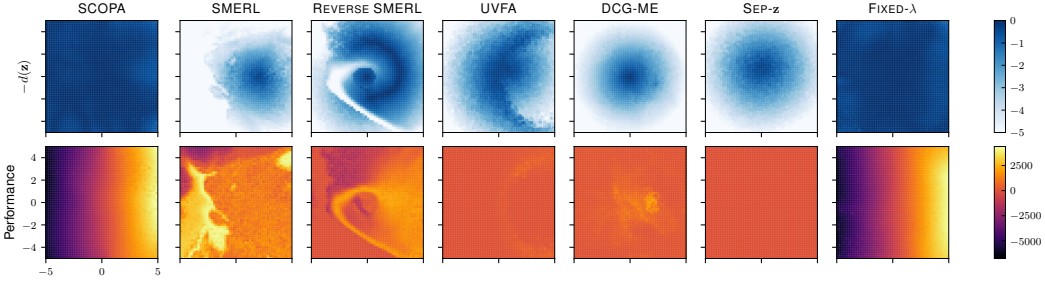

Figure A.20: **Ant Velocity** Heatmaps of **(top)** negative distance to skill, **(bottom)** performance defined in Section 5.3 for all baselines. In the bottom row, the skills that are not successfully executed (i.e. $d(\mathbf{z}) > d_{\text{eval}}$) are **not** filtered out.

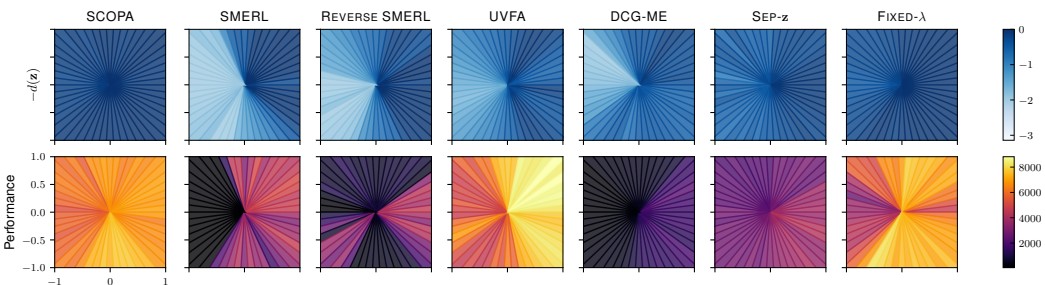

Figure A.21: **Humanoid Angle** Heatmaps of **(top)** negative distance to skill, **(bottom)** performance defined in Section 5.3 for all baselines. In the bottom row, the skills that are not successfully executed (i.e. $d(\mathbf{z}) > d_{\text{eval}}$) are **not** filtered out.

## B   THEORETICAL RESULTS

**Proposition.** *Consider an infinite horizon, finite MDP with observable features in $\Phi$. Let $\pi$ be a policy and let $\psi$ be the discounted successor features. Then, for all skills $\mathbf{z} \in \mathcal{Z}$, we can derive an upper bound for the distance between $\mathbf{z}$ and the expected features under $\pi$:*

$$\left\| \mathbb{E}_{\pi(\cdot|\cdot,\mathbf{z})} [\phi(s)] - \mathbf{z} \right\|_2 \leq \mathbb{E}_{\pi(\cdot|\cdot,\mathbf{z})} \left[ \| (1-\gamma)\psi(s,\mathbf{z}) - \mathbf{z} \|_2 \right] \tag{5}$$

*Proof.* For all states $s \in \mathcal{S}$, the Bellman equation for $\psi$ gives:

$$\psi(s,\mathbf{z}) = \phi(s) + \gamma \mathbb{E}_{\substack{a \sim \pi(\cdot|s,\mathbf{z}) \\ s' \sim p(\cdot|s,a)}} [\psi(s',\mathbf{z})] \tag{6}$$

For all skills $\mathbf{z} \in \mathcal{Z}$ and for all sequences of $T$ states $(s_0, \ldots, s_{T-1})$ sampled from $\pi(\cdot|\cdot, \mathbf{z})$, we have:

$$\left\| \frac{1}{T} \sum_{t=0}^{T-1} \phi_t - \mathbf{z} \right\|_2$$

$$= \left\| \frac{1}{T} \sum_{t=0}^{T-1} (\phi_t - \mathbf{z}) \right\|_2$$

$$= \left\| \frac{1}{T} \sum_{t=0}^{T-1} \left( \psi(s_t,\mathbf{z}) - \gamma \mathbb{E}_{\substack{a_t \sim \pi(\cdot|s_t,\mathbf{z}) \\ s_{t+1} \sim p(\cdot|s_t,a_t)}} [\psi(s_{t+1},\mathbf{z})] - \mathbf{z} \right) \right\|_2 \qquad \text{(Equation 6)}$$

$$= \left\| \frac{1}{T} \sum_{t=0}^{T-1} \left( (1-\gamma)\psi(s_t,\mathbf{z}) + \gamma\psi(s_t,\mathbf{z}) - \gamma \mathbb{E}_{\substack{a_t \sim \pi(\cdot|s_t,\mathbf{z}) \\ s_{t+1} \sim p(\cdot|s_t,a_t)}} [\psi(s_{t+1},\mathbf{z})] - \mathbf{z} \right) \right\|_2$$

$$= \left\| \frac{1}{T} \sum_{t=0}^{T-1} ((1-\gamma)\psi(s_t,\mathbf{z}) - \mathbf{z}) + \frac{\gamma}{T} \sum_{t=0}^{T-1} \left( \psi(s_t,\mathbf{z}) - \mathbb{E}_{\substack{a_t \sim \pi(\cdot|s_t,\mathbf{z}) \\ s_{t+1} \sim p(\cdot|s_t,a_t)}} [\psi(s_{t+1},\mathbf{z})] \right) \right\|_2$$

$$\leq \frac{1}{T} \sum_{t=0}^{T-1} \| (1-\gamma)\psi(s_t,\mathbf{z}) - \mathbf{z} \|_2 + \gamma \left\| \frac{1}{T} \sum_{t=0}^{T-1} \left( \psi(s_t,\mathbf{z}) - \mathbb{E}_{\substack{a_t \sim \pi(\cdot|s_t,\mathbf{z}) \\ s_{t+1} \sim p(\cdot|s_t,a_t)}} [\psi(s_{t+1},\mathbf{z})] \right) \right\|_2$$

$$\text{(triangular inequality)}$$

We denote $\rho(s) = \lim_{t \to \infty} P(s_t = s | s_0, \pi)$ the stationary distribution of states under $\pi$, which we assume exists and is independent of $s_0$. Thus, we can write:

$$\lim_{T \to \infty} \frac{1}{T} \sum_{t=0}^{T-1} \left( \psi(s_t,\mathbf{z}) - \mathbb{E}_{\substack{a_t \sim \pi(\cdot|s_t,\mathbf{z}) \\ s_{t+1} \sim p(\cdot|s_t,a_t)}} [\psi(s_{t+1},\mathbf{z})] \right)$$

$$= \lim_{T \to \infty} \frac{1}{T} \sum_{t=0}^{T-1} \psi(s_t,\mathbf{z}) - \lim_{T \to \infty} \frac{1}{T} \sum_{t=0}^{T-1} \left( \mathbb{E}_{\substack{a_t \sim \pi(\cdot|s_t,\mathbf{z}) \\ s_{t+1} \sim p(\cdot|s_t,a_t)}} [\psi(s_{t+1},\mathbf{z})] \right)$$

$$= \mathbb{E}_{s \sim \rho} [\psi(s,\mathbf{z})] - \mathbb{E}_{s \sim \rho} \left[ \mathbb{E}_{\substack{a \sim \pi(\cdot|s,\mathbf{z}) \\ s' \sim p(\cdot|s,a)}} [\psi(s',\mathbf{z})] \right]$$

$$= \mathbb{E}_{s \sim \rho} [\psi(s,\mathbf{z})] - \mathbb{E}_{\substack{s \sim \rho \\ a \sim \pi(\cdot|s,\mathbf{z}) \\ s' \sim p(\cdot|s,a)}} [\psi(s',\mathbf{z})]$$

$$= \mathbb{E}_{s \sim \rho} [\psi(s,\mathbf{z})] - \mathbb{E}_{s' \sim \rho} [\psi(s',\mathbf{z})]$$

$$= 0$$

Finally, by taking the inequality to the limit as $T \to \infty$, we get:

$$\left\| \mathbb{E}_{\pi(\cdot|\cdot,\mathbf{z})} [\phi(s)] - \mathbf{z} \right\|_2 \leq \mathbb{E}_{\pi(\cdot|\cdot,\mathbf{z})} \left[ \| (1-\gamma)\psi(s,\mathbf{z}) - \mathbf{z} \|_2 \right] + 0$$

$$\boxed{\left\| \mathbb{E}_{\pi(\cdot|\cdot,\mathbf{z})} [\phi(s)] - \mathbf{z} \right\|_2 \leq \mathbb{E}_{\pi(\cdot|\cdot,\mathbf{z})} \left[ \| (1-\gamma)\psi(s,\mathbf{z}) - \mathbf{z} \|_2 \right]}$$

$\square$

**Proposition B.1.** *Consider a continuous MDP with a bounded feature space* $\Phi$, *a skill* $\mathbf{z} \in \mathcal{Z}$, *and* $\pi$ *a policy such that the sequence* $\left( \frac{1}{T} \sum_{t=0}^{T-1} \phi_t \right)_{T \geq 1}$ *almost surely[1] converges for trajectories sampled from* $\pi(\cdot | \cdot, \mathbf{z})$. *If we write* $\epsilon := \sup_t \mathbb{E}_{\pi(\cdot|\cdot,\mathbf{z})} [\|\phi_t + \gamma \psi(s_t, \mathbf{z}) - \psi(s_{t+1}, \mathbf{z})\|_2]$, *then:*

$$\mathbb{E}_{\pi(\cdot|\cdot,\mathbf{z})} \left[ \left\| \lim_{T \to \infty} \frac{1}{T} \sum_{t=0}^{T-1} \phi_t - \mathbf{z} \right\|_2 \right] \leq \sup_t \mathbb{E}_{\pi(\cdot|\cdot,\mathbf{z})} [\|(1-\gamma)\psi(s_t, \mathbf{z}) - \mathbf{z}\|_2] + \epsilon \quad (7)$$

*Furthermore, it is worth noting that if the MDP dynamics* $p$ *and* $\pi$ *are deterministic, then* $\epsilon = 0$.

*Proof.* Let $\mathbf{z} \in \mathcal{Z}$.

To make the proof easier to read, we use the following notations:

$$\psi_t := \psi(s_t, \mathbf{z})$$
$$\pi_{\mathbf{z}} := \pi(\cdot | \cdot, \mathbf{z})$$

We define $\beta$ as follows:

$$\beta := \sup_t \mathbb{E}_{\pi_{\mathbf{z}}} [\|(1-\gamma)\psi_t - \mathbf{z}\|_2] \quad (8)$$

Then we have, for all $t$,

$$\mathbb{E}_{\pi_{\mathbf{z}}} [\|(1-\gamma)\psi_t - \mathbf{z}\|_2] \leq \beta \quad (9)$$
$$\mathbb{E}_{\pi_{\mathbf{z}}} [\|\phi_t + \gamma\psi_t - \psi_{t+1}\|_2] \leq \epsilon \quad (10)$$

The Bellman equation applied to Successor Features (SF) can be written:

$$\psi_t = \phi_t + \gamma \mathbb{E}_{\pi_{\mathbf{z}}} [\psi_{t+1} | s_t] \quad (11)$$
$$\text{or also: } \phi_t = \psi_t - \gamma \mathbb{E}_{\pi_{\mathbf{z}}} [\psi_{t+1} | s_t] \quad (12)$$

We can now derive an upper bound for $\left\| \frac{1}{T} \sum_{t=0}^{T-1} \phi_t - \mathbf{z} \right\|_2$. For all sequences of $T$ states $s_{0:T-1}$ we have:

$$\left\| \frac{1}{T} \sum_{t=0}^{T-1} \phi_t - \mathbf{z} \right\|_2 = \left\| \frac{1}{T} \sum_{t=0}^{T-1} (\phi_t - \mathbf{z}) \right\|_2$$

$$= \left\| \frac{1}{T} \sum_{t=0}^{T-1} \left( \psi_t - \gamma \mathbb{E}_{\pi_{\mathbf{z}}} [\psi_{t+1} | s_t] - \mathbf{z} \right) \right\|_2 \quad \text{(from Equation 12)}$$

$$= \left\| \frac{1}{T} \sum_{t=0}^{T-1} \left( (1-\gamma)\psi_t + \gamma\psi_t - \gamma \mathbb{E}_{\pi_{\mathbf{z}}} [\psi_{t+1} | s_t] - \mathbf{z} \right) \right\|_2$$

$$= \left\| \frac{1}{T} \sum_{t=0}^{T-1} ((1-\gamma)\psi_t - \mathbf{z}) + \frac{1}{T} \sum_{t=0}^{T-1} \left( \gamma\psi_t - \gamma \mathbb{E}_{\pi_{\mathbf{z}}} [\psi_{t+1} | s_t] \right) \right\|_2$$

$$\leq \left\| \frac{1}{T} \sum_{t=0}^{T-1} ((1-\gamma)\psi_t - \mathbf{z}) \right\|_2 + \left\| \frac{1}{T} \sum_{t=0}^{T-1} \left( \gamma\psi_t - \gamma \mathbb{E}_{\pi_{\mathbf{z}}} [\psi_{t+1} | s_t] \right) \right\|_2$$
$$\text{(triangular inequality)}$$

Thus,

$$\mathbb{E}_{\pi_{\mathbf{z}}} \left[ \left\| \frac{1}{T} \sum_{t=0}^{T-1} \phi_t - \mathbf{z} \right\|_2 \right] \leq \mathbb{E}_{\pi_{\mathbf{z}}} \left[ \left\| \frac{1}{T} \sum_{t=0}^{T-1} ((1-\gamma)\psi_t - \mathbf{z}) \right\|_2 \right]$$
$$+ \mathbb{E}_{\pi_{\mathbf{z}}} \left[ \left\| \frac{1}{T} \sum_{t=0}^{T-1} \left( \gamma\psi_t - \gamma \mathbb{E}_{\pi_{\mathbf{z}}} [\psi_{t+1} | s_t] \right) \right\|_2 \right] \quad (13)$$

---

[1] *almost sure* refers to the almost sure convergence from probability theory where rollouts are sampled from $\pi(\cdot | \cdot, \mathbf{z})$.

We consider now the two terms on the right hand-side separately. First of all, we prove that the first term is lower than or equal to $\beta$:

$$\mathbb{E}_{\pi_{\mathbf{z}}}\left[\left\|\frac{1}{T}\sum_{t=0}^{T-1}((1-\gamma)\boldsymbol{\psi}_t-\mathbf{z})\right\|_2\right] \leq \mathbb{E}_{\pi_{\mathbf{z}}}\left[\frac{1}{T}\sum_{t=0}^{T-1}\|((1-\gamma)\boldsymbol{\psi}_t-\mathbf{z})\|_2\right] \quad \text{(triangular inequality)}$$

$$\leq \frac{1}{T}\sum_{t=0}^{T-1}\mathbb{E}_{\pi_{\mathbf{z}}}\left[\|((1-\gamma)\boldsymbol{\psi}_t-\mathbf{z})\|_2\right]$$

$$\leq \frac{1}{T}\sum_{t=0}^{T-1}\beta \quad \text{(from Equation 9)}$$

$$\leq \beta \quad (14)$$

Also, we can prove that the second term of the right-hand side in Equation 13 is lower than or equal to $\epsilon + \eta_T$ where $\lim_{T\to\infty}\eta_T = 0$. For all sequences of $T$ states $s_{0:T-1}$, we have:

$$\sum_{t=0}^{T-1}\left(\gamma\boldsymbol{\psi}_t - \gamma\mathbb{E}_{\pi_{\mathbf{z}}}\left[\boldsymbol{\psi}_{t+1}\big|s_t\right]\right) = \sum_{t=0}^{T-1}\gamma\boldsymbol{\psi}_t - \sum_{t=0}^{T-1}\gamma\mathbb{E}_{\pi_{\mathbf{z}}}\left[\boldsymbol{\psi}_{t+1}\big|s_t\right]$$

$$= \gamma\boldsymbol{\psi}_0 - \gamma\mathbb{E}_{\pi_{\mathbf{z}}}\left[\boldsymbol{\psi}_T\big|s_{T-1}\right] + \sum_{t=1}^{T-1}\gamma\boldsymbol{\psi}_t - \sum_{t=0}^{T-2}\gamma\mathbb{E}_{\pi_{\mathbf{z}}}\left[\boldsymbol{\psi}_{t+1}\big|s_t\right]$$

$$= \gamma\boldsymbol{\psi}_0 - \gamma\mathbb{E}_{\pi_{\mathbf{z}}}\left[\boldsymbol{\psi}_T\big|s_{T-1}\right] + \sum_{t=0}^{T-2}\gamma\boldsymbol{\psi}_{t+1} - \sum_{t=0}^{T-2}\gamma\mathbb{E}_{\pi_{\mathbf{z}}}\left[\boldsymbol{\psi}_{t+1}\big|s_t\right]$$

$$= \gamma\boldsymbol{\psi}_0 - \gamma\mathbb{E}_{\pi_{\mathbf{z}}}\left[\boldsymbol{\psi}_T\big|s_{T-1}\right] + \sum_{t=0}^{T-2}\left(\gamma\boldsymbol{\psi}_{t+1} - \gamma\mathbb{E}_{\pi_{\mathbf{z}}}\left[\boldsymbol{\psi}_{t+1}\big|s_t\right]\right)$$

Thus, after dividing by $T$ and applying the norm and expectation, we get:

$$\mathbb{E}_{\pi_{\mathbf{z}}}\left[\left\|\frac{1}{T}\sum_{t=0}^{T-1}\left(\gamma\boldsymbol{\psi}_t - \gamma\mathbb{E}_{\pi_{\mathbf{z}}}\left[\boldsymbol{\psi}_{t+1}\big|s_t\right]\right)\right\|_2\right] \leq \mathbb{E}_{\pi_{\mathbf{z}}}\left[\left\|\frac{1}{T}\left(\gamma\boldsymbol{\psi}_0 - \gamma\mathbb{E}_{\pi_{\mathbf{z}}}\left[\boldsymbol{\psi}_T\big|s_{T-1}\right]\right)\right\|_2\right]$$

$$+ \mathbb{E}_{\pi_{\mathbf{z}}}\left[\left\|\frac{1}{T}\sum_{t=0}^{T-2}\left(\gamma\boldsymbol{\psi}_{t+1} - \gamma\mathbb{E}_{\pi_{\mathbf{z}}}\left[\boldsymbol{\psi}_{t+1}\big|s_t\right]\right)\right\|_2\right]$$

$$\text{(triangular inequality)}$$

Let $\eta_T := \mathbb{E}_{\pi_{\mathbf{z}}} \left[ \left\| \frac{1}{T} \left( \gamma \boldsymbol{\psi}_0 - \gamma \mathbb{E}_{\pi_{\mathbf{z}}} \left[ \boldsymbol{\psi}_T \middle| s_{T-1} \right] \right) \right\|_2 \right]$, we then have:

$$
\mathbb{E}_{\pi_{\mathbf{z}}} \left[ \left\| \frac{1}{T} \sum_{t=0}^{T-1} \left( \gamma \boldsymbol{\psi}_t - \gamma \mathbb{E}_{\pi_{\mathbf{z}}} \left[ \boldsymbol{\psi}_{t+1} \middle| s_t \right] \right) \right\|_2 \right]
$$

$$
\leq \eta_T + \mathbb{E}_{\pi_{\mathbf{z}}} \left[ \left\| \frac{1}{T} \sum_{t=0}^{T-2} \left( \gamma \boldsymbol{\psi}_{t+1} - \gamma \mathbb{E}_{\pi_{\mathbf{z}}} \left[ \boldsymbol{\psi}_{t+1} \middle| s_t \right] \right) \right\|_2 \right] \quad \text{(triangular inequality)}
$$

$$
\leq \eta_T + \mathbb{E}_{\pi_{\mathbf{z}}} \left[ \frac{1}{T} \sum_{t=0}^{T-2} \left\| \gamma \boldsymbol{\psi}_{t+1} - \gamma \mathbb{E}_{\pi_{\mathbf{z}}} \left[ \boldsymbol{\psi}_{t+1} \middle| s_t \right] \right\|_2 \right]
$$

$$
\leq \eta_T + \mathbb{E}_{\pi_{\mathbf{z}}} \left[ \frac{1}{T} \sum_{t=0}^{T-2} \left\| \gamma \boldsymbol{\psi}_{t+1} + \boldsymbol{\phi}_t - \boldsymbol{\psi}_t \right\|_2 \right] \quad \text{(from Equation 11)}
$$

$$
\leq \eta_T + \frac{1}{T} \sum_{t=0}^{T-2} \mathbb{E}_{\pi_{\mathbf{z}}} \left[ \left\| \gamma \boldsymbol{\psi}_{t+1} + \boldsymbol{\phi}_t - \boldsymbol{\psi}_t \right\|_2 \right]
$$

$$
\leq \eta_T + \frac{1}{T} \sum_{t=0}^{T-2} \mathbb{E}_{\pi_{\mathbf{z}}} \left[ \left\| \boldsymbol{\phi}_t + \gamma \boldsymbol{\psi}_{t+1} - \boldsymbol{\psi}_t \right\|_2 \right]
$$

$$
\leq \eta_T + \frac{1}{T} \sum_{t=0}^{T-2} \epsilon
$$

$$
\leq \eta_T + \frac{T-1}{T} \epsilon \tag{15}
$$

After combining the two previously derived Equations 14 and 15, we get:

$$
\mathbb{E}_{\pi_{\mathbf{z}}} \left[ \left\| \frac{1}{T} \sum_{t=0}^{T-1} \boldsymbol{\phi}_t - \mathbf{z} \right\|_2 \right] \leq \beta + \eta_T + \frac{T-1}{T} \epsilon \tag{16}
$$

Now we intend to prove that $\lim_{T \to \infty} \eta_T = 0$

$$
\eta_T = \mathbb{E}_{\pi_{\mathbf{z}}} \left[ \left\| \frac{1}{T} \left( \gamma \boldsymbol{\psi}_0 - \gamma \mathbb{E}_{\pi_{\mathbf{z}}} \left[ \boldsymbol{\psi}_T \middle| s_{T-1} \right] \right) \right\|_2 \right]
$$

$$
= \frac{\gamma}{T} \mathbb{E}_{\pi_{\mathbf{z}}} \left[ \left\| \left( \boldsymbol{\psi}_0 - \mathbb{E}_{\pi_{\mathbf{z}}} \left[ \boldsymbol{\psi}_T \middle| s_{T-1} \right] \right) \right\|_2 \right]
$$

$$
\leq \frac{\gamma}{T} \mathbb{E}_{\pi_{\mathbf{z}}} \left[ \left\| \boldsymbol{\psi}_0 \right\|_2 + \left\| \mathbb{E}_{\pi_{\mathbf{z}}} \left[ \boldsymbol{\psi}_T \middle| s_{T-1} \right] \right\|_2 \right] \quad \text{(triangular inequality)}
$$

$$
\leq \frac{\gamma}{T} \left( \mathbb{E}_{\pi_{\mathbf{z}}} \left[ \left\| \boldsymbol{\psi}_0 \right\|_2 \right] + \mathbb{E}_{\pi_{\mathbf{z}}} \left[ \left\| \mathbb{E}_{\pi_{\mathbf{z}}} \left[ \boldsymbol{\psi}_T \middle| s_{T-1} \right] \right\|_2 \right] \right)
$$

As the space of features $\Phi$ is bounded, there exist a $\rho > 0$ such that for all $\phi \in \Phi$, $\|\phi\|_2 \leq \rho$. Hence, for all $t$, $\|\boldsymbol{\psi}_t\|_2 = \mathbb{E}_{\pi_{\mathbf{z}}} \left[ \left\| \sum_{i=0}^{\infty} \gamma^i \boldsymbol{\phi}_{t+i} \right\|_2 \middle| s_t \right] \leq \mathbb{E}_{\pi_{\mathbf{z}}} \left[ \sum_{i=0}^{\infty} \gamma^i \left\| \boldsymbol{\phi}_{t+i} \right\|_2 \middle| s_t \right] \leq \frac{\rho}{1-\gamma}$. Hence,

$$
\eta_T \leq \frac{\gamma}{T} \left( \frac{\rho}{1-\gamma} + \mathbb{E}_{\pi_{\mathbf{z}}} \left[ \left\| \mathbb{E}_{\pi_{\mathbf{z}}} \left[ \boldsymbol{\psi}_T \middle| s_{T-1} \right] \right\|_2 \right] \right)
$$

$$
\leq \frac{\gamma}{T} \left( \frac{\rho}{1-\gamma} + \mathbb{E}_{\pi_{\mathbf{z}}} \left[ \mathbb{E}_{\pi_{\mathbf{z}}} \left[ \left\| \boldsymbol{\psi}_T \right\|_2 \middle| s_{T-1} \right] \right] \right) \quad \text{(Jensen's inequality)}
$$

$$
\leq \frac{\gamma}{T} \left( \frac{\rho}{1-\gamma} + \mathbb{E}_{\pi_{\mathbf{z}}} \left[ \left\| \boldsymbol{\psi}_T \right\|_2 \right] \right) \quad \text{(law of total expectation)}
$$

$$
\leq \frac{\gamma}{T} \left( \frac{\rho}{1-\gamma} + \frac{\rho}{1-\gamma} \right)
$$

$$
\leq \frac{1}{T} \left( \frac{2\rho\gamma}{1-\gamma} \right) \tag{17}
$$

Then, knowing that for all $T$, we have $0 \leq \eta_T$, the squeeze theorem ensures that $\lim_{T \to \infty} \eta_T = 0$.

Now we will prove that the left-hand side of Equation 13 converges, let $X_T := \left\| \frac{1}{T} \sum_{t=0}^{T-1} \phi_t - \mathbf{z} \right\|_2$. For all $T$,

$$
\begin{aligned}
|X_T| &\leq \frac{1}{T} \sum_{t=0}^{T-1} \|\phi_t\|_2 + \|\mathbf{z}\|_2 \qquad \text{(triangular inequality)} \\
&\leq \rho + \|\mathbf{z}\|_2
\end{aligned}
$$

Moreover, $\mathbf{z}$ is a fixed variable, which means that $|X_T|$ is bounded. In addition, $\mathbb{E}_{\pi_{\mathbf{z}}} [\rho + \|\mathbf{z}\|_2] < \infty$, and the sequence $(X_T)_{T \geq 1}$ converges almost surely (since $\left( \frac{1}{T} \sum_{t=0}^{T-1} \phi_t \right)_{T \geq 1}$ converges almost surely by hypothesis). The dominated convergence theorem then ensures that $(\mathbb{E}_{\pi_{\mathbf{z}}} [X_T])_{T \geq 1}$ converges and:

$$
\lim_{T \to \infty} \mathbb{E}_{\pi_{\mathbf{z}}} [X_T] = \mathbb{E}_{\pi_{\mathbf{z}}} \left[ \lim_{T \to \infty} X_T \right] \tag{18}
$$

Finally, by taking the Equation 13 to the limit as $T \to \infty$, we get:

$$
\lim_{T \to \infty} \mathbb{E}_{\pi_{\mathbf{z}}} \left[ \underbrace{\left\| \frac{1}{T} \sum_{t=0}^{T-1} \phi_t - \mathbf{z} \right\|_2}_{X_T} \right] \leq \lim_{T \to \infty} \left( \beta + \eta_T + \frac{T-1}{T} \epsilon \right)
$$

$$
\mathbb{E}_{\pi_{\mathbf{z}}} \left[ \left\| \lim_{T \to \infty} \frac{1}{T} \sum_{t=0}^{T-1} \phi_t - \mathbf{z} \right\|_2 \right] \leq \beta + \underbrace{\lim_{T \to \infty} \eta_T}_{=0} + \underbrace{\lim_{T \to \infty} \frac{T-1}{T}}_{=1} \epsilon
$$

$$
\boxed{\mathbb{E}_{\pi_{\mathbf{z}}} \left[ \left\| \lim_{T \to \infty} \frac{1}{T} \sum_{t=0}^{T-1} \phi_t - \mathbf{z} \right\|_2 \right] \leq \beta + \epsilon}
$$

$\square$

# C    TASKS DETAILS

## C.1    TASKS

Table C.1: Tasks

| | FEET CONTACT | | | JUMP | VELOCITY | ANGLE |
|---|---|---|---|---|---|---|
| | HUMANOID | ANT | WALKER | HUMANOID | ANT | HUMANOID |
| STATE DIM | 244 | 27 | 17 | 244 | 27 | 244 |
| ACTION DIM | 17 | 8 | 6 | 17 | 8 | 17 |
| FEATURES DIM | 2 | 4 | 2 | 1 | 1 | 1 |
| FEATURES SPACE | $\{0,1\}^2$ | $\{0,1\}^4$ | $\{0,1\}^2$ | $[0,0.25]$ | $[-5.,5]^2$ | $]-\pi,\pi]$ |
| SKILL SPACE | $[0,1]^2$ | $[0,1]^4$ | $[0,1]^2$ | $[0,0.25]$ | $[-5.,5]^2$ | $]-\pi,\pi]$ |
| EPISODE LENGTH | 1000 | 1000 | 1000 | 1000 | 1000 | 1000 |
| THRESHOLD $\delta$ | 0.01 | 0.1 | 0.01 | 0.0025 | 0.1 | 0.06 |
| DISTANCE EVAL $d_{\text{eval}}$ | 0.05 | 0.5 | 0.05 | 0.0125 | 0.5 | 0.30 |

## C.2 ADAPTATION TASKS

For all adaptation tasks, the reward stays the same but the dynamics of the MDP is changed. The goal is to leverage the diversity of skills to adapt to unforeseen situations.

Table C.2: Adaptation tasks

| | HURDLES HUMANOID | MOTOR FAILURE HUMANOID | GRAVITY HUMANOID | FRICTION WALKER | OBSTACLE ANT |
|---|---|---|---|---|---|
| FEATURES ADAPTATION | Jump Few-shot | Feet Contact Few-shot | Feet Contact Few-shot | Feet Contact Few-shot | Velocity Hierarchy |

### C.2.1 FEW SHOT ADAPTATION

For all few-shot adaptation tasks, we evaluate all skills for each replication of each method and select the best one to solve the adaptation task. In Figure 5, the lines represent the IQM for the 10 replications and the shaded areas correspond to the 95% CI.

On **Humanoid - Hurdles**, we use the jump skills to jump over hurdles varying in height from 0 to 50 cm.

On **Humanoid - Motor Failure**, we use the feet contact skills to find the best way to continue walking forward despite the damage. In this experiment, we scale the action corresponding to the torque of the left knee (actuator 10) by the damage strength (x-axis of Figure 5) ranging from 0.0 (no damage) to 1.0 (maximal damage).

On **Ant - Gravity**, we use the feet contact skills to find the best way to continue walking forward despite the change in gravity. In this experiment, we scale the gravity by a coefficient ranging from 0.5 (low gravity) to 3.0 (high gravity).

On **Walker - Friction**, we use the feet contact skills to find the best way to continue walking forward despite the change in friction. In this experiment, we scale the friction by a coefficient ranging from 0.0 (low friction) to 5.0 (high friction).

### C.2.2 HIERARCHICAL LEARNING

For the hierarchical learning task, we use each replication of each method to learn a meta-controller that selects the skills of the policy in order to adapt to the new task.

On **Ant - Obstacle**, the meta-controller is trained with SAC to select the velocity skills that enables to go around the obstacle and move forward as fast as possible in order to maximize performance.

# D    BASELINES DETAILS

Table D.3: Comparison of the main features for the different algorithms

| Algorithm | Objective Function | Model-based | Lagrange Multiplier $\lambda$ |
|-----------|-------------------|-------------|-------------------------------|
| SCOPA | $(1-\lambda)\sum\gamma^t r_t - \lambda\left\|(1-\gamma)\psi - \mathbf{z}\right\|_2$ | ✓ | Learned, function of $\mathbf{z}$ |
| SMERL | $\sum\gamma^t\left(r_t - \lambda\mathbb{1}(R \geq R^* - \epsilon)\left\|\mu(\phi_t) - \mathbf{z}_{\text{DIAYN}}\right\|_2^2\right)^*$ | ✗ | Fixed |
| REVERSE SMERL | $\sum\gamma^t\left(\mathbb{1}(R < R^* - \epsilon)r_t - \lambda\left\|\mu(\phi_t) - \mathbf{z}_{\text{DIAYN}}\right\|_2^2\right)^*$ | ✗ | Fixed |
| DCG-ME ** | $\sum\gamma^t r_t$ | ✗ | — |
| UVFA | $(1-\lambda)\sum\gamma^t r_t - \lambda\sum\gamma^t\left\|\phi_t - \mathbf{z}\right\|_2$ | ✓ | Fixed |
| SEP-$\mathbf{z}$ | $(1-\lambda)\sum\gamma^t r_t - \lambda\sum\gamma^t\left\|\phi_t - \mathbf{z}\right\|_2$ | ✓ | Learned, function of $\mathbf{z}$ |
| FIXED-$\lambda$ | $(1-\lambda)\sum\gamma^t r_t - \lambda\left\|(1-\gamma)\psi - \mathbf{z}\right\|_2$ | ✓ | Fixed |

\* see Section D.1 for a detailed explanation of the reward used in SMERL and REVERSE SMERL.
\*\* DCG-ME learns diverse skills with a mechanism that is not visible in its objective function.

## D.1    SMERL

To estimate the optimal return $R_{\mathcal{M}}(\pi_{\mathcal{M}}^*)$ required by SMERL, we apply the same method as Kumar et al. (2020). We trained SAC on each environment and used SAC performance $R_{\text{SAC}}$ as the the optimal return value for each environment. Similarly to Kumar et al. (2020), we choose $\lambda = 2.0$ by taking the best value when evaluated on HalfCheetah environment.

We use SMERL with continuous DIAYN with a Gaussian discriminator (Choi et al., 2021), so that the policy learns a continuous range of skills instead of a finite number of skills (Kumar et al., 2020). We also use DIAYN+prior (Eysenbach et al., 2018; Chalumeau et al., 2022) with $q(\mathbf{z}_{\text{DIAYN}}|s)$ and feature priors to guide SMERL towards relevant skills as explained in DIAYN original paper.

With a Gaussian discriminator $q(\mathbf{z}_{\text{DIAYN}}|s) = \mathcal{N}(\mathbf{z}_{\text{DIAYN}}|\mu(s), \Sigma(s))$, the intrinsic reward is of the form $\tilde{r} = \log q(\mathbf{z}_{\text{DIAYN}}|s) - \log p(\mathbf{z}_{\text{DIAYN}}) \propto \left\|\mu(s) - \mathbf{z}_{\text{DIAYN}}\right\|_2^2$ up to an additive and a multiplicative constant, as demonstrated by Choi et al. (2021). Replacing the state $s$ with the prior information $\phi(s)$ in the discriminator gives $\tilde{r} \propto -\left\|\mu(\phi(s)) - \mathbf{z}_{\text{DIAYN}}\right\|_2^2$. Consequently, we can see that the intrinsic reward from DIAYN corresponds to executing a latent skill $\mathbf{z}_{\text{DIAYN}}$ (i.e. achieving a latent goal) in the unsupervised space defined by the discriminator $q(\mathbf{z}_{\text{DIAYN}}|\phi(s))$. Indeed, the intrinsic reward is analogous to the reward used in GCRL of the form $r \propto -\left\|\phi(s) - \mathbf{g}\right\|_2$ (Liu et al., 2022). Moreover, the bijection between the latent skills (i.e. latent goals) and the features (i.e. goals) is given by $\mathbf{z}_{\text{DIAYN}} \sim q(\mathbf{z}_{\text{DIAYN}}|\phi(s))$.

Finally, we explain how we can use the previous analysis of DIAYN's intrinsic reward to calculate the distance to skill metric. Given a skill $\mathbf{z}$, we can use the discriminator to get the Maximum Likelihood (ML) of the distribution $\mathbf{z}_{\text{DIAYN}} = \text{ML}(q(\mathbf{z}_{\text{DIAYN}}|\mathbf{z})) = \text{ML}(\mathcal{N}(\mu(\mathbf{z}), \Sigma(\mathbf{z}))) = \mu(\mathbf{z})$. Given $\mathbf{z}_{\text{DIAYN}}$, we can rollout the policy $\pi(.|., \mathbf{z}_{\text{DIAYN}})$ in the environment and get a trajectory. From the trajectory, we can observe the achieved skill $\mathbf{z}_{\text{achieved}}$, and then calculate the distance to skill as $\left\|\mathbf{z} - \mathbf{z}_{\text{achieved}}\right\|_2$.

## D.2    REVERSE SMERL

For REVERSE SMERL, we use exactly the same hyperparameters as SMERL.

The same discussion holds for SMERL and REVERSE SMERL holds, see Appendix D.1 for more details.

## D.3    SEP-$\mathbf{z}$

We can show that the constraint in SCOPA's objective function is easier to satisfy than SEP-$\mathbf{z}$'s constraint.

For all skills $\mathbf{z} \in \mathcal{Z}$ and for all sequences of states $(s_t)_{t \geq 0}$, we have:

$$
\begin{aligned}
\left\| (1-\gamma) \sum_{t=0}^{\infty} \gamma^t \phi_t - \mathbf{z} \right\|_2 &= \left\| (1-\gamma) \left( \sum_{t=0}^{\infty} \gamma^t \phi_t - \sum_{t=0}^{\infty} \gamma^t \mathbf{z} \right) \right\|_2 \\
&= \left\| (1-\gamma) \sum_{t=0}^{\infty} \gamma^t (\phi_t - \mathbf{z}) \right\|_2 \\
&\leq (1-\gamma) \sum_{t=0}^{\infty} \gamma^t \left\| \phi_t - \mathbf{z} \right\|_2
\end{aligned}
$$

Thus, we have the following inequality:

$$
\| (1-\gamma) \psi(s, \mathbf{z}) - \mathbf{z} \|_2 \leq (1-\gamma) \sum_{t=0}^{\infty} \gamma^t \left\| \phi_t - \mathbf{z} \right\|_2
$$

At each timestep $t$, SEP-$\mathbf{z}$ tries to satisfy $\phi_t = \mathbf{z}$, whereas SCOPA approximately tries to satisfy $\lim_{T \to \infty} \frac{1}{T} \sum_{t=0}^{T} \phi_t = \mathbf{z}$, which is less restrictive.

# E  HYPERPARAMETERS

We provide here all the hyperparameters used for the baselines and SCOPA. SCOPA uses the same hyperparameters as the ones used by DREAMERV3 (Hafner et al., 2023), hence we provide here only the parameters mentioned in this work.

Our implementation is based on the implementation of DREAMERV3. Our successor features network is also implemented as a distributional critic. In our implementation, the Lagrangian network $\lambda_{\mathcal{L}}$ is only conditioned on the skill $\mathbf{z}$, as we noticed no difference in performance.

The hyperparameters for SMERL, REVERSE SMERL and DCG-ME are exactly the same as in their original papers, where they were fine-tuned on similar locomotion tasks. We also tried using hidden layers of size 512 for those baselines, in order to give them an architecture that is closer to SCOPA. We noticed this deteriorates the results of Reverse SMERL and DCG-ME, and this does not significantly increase (statistically speaking) the performance of SMERL.

Each algorithm is run for $10^7$ environment steps, to ensure that they have enough iterations to converge.

Table E.4: SCOPA hyperparameters

| Parameter | Value |
| --- | --- |
| Actor network | $[512, 512, |\mathcal{A}|]$ |
| Critic network | $[512, 512, 1]$ |
| Successor Feature network | $[512, 512, |\mathcal{Z}|]$ |
| Lagrangian network | $[8, 1]$ |
| Imagination batch size $N$ | 1024 |
| Real environment exploration batch size | 16 |
| Total timesteps | $1 \times 10^7$ |
| Optimizer | Adam |
| Learning rate | $3 \times 10^{-4}$ |
| Replay buffer size | $10^6$ |
| Discount factor $\gamma$ | 0.997 |
| Imagination horizon $H$ | 15 |
| Target smoothing coefficient $\tau$ | 0.005 |
| Sampling period $T$ | 100 |
| Lambda Return $\lambda$ | 0.95 |

Table E.5: SMERL hyperparameters

| Parameter | Value |
| --- | --- |
| Actor network | $[256, 256, |\mathcal{A}|]$ |
| Critic network | $[256, 256, 1]$ |
| Batch size | 256 |
| Optimizer | Adam |
| Learning rate | $3 \times 10^{-4}$ |
| Replay buffer size | $10^6$ |
| Discount factor $\gamma$ | 0.99 |
| Target smoothing coefficient $\tau$ | 0.005 |
| Skill distribution | Normal distribution |
| Diversity reward scale | 10.0 |
| SMERL target | $R_{\text{SAC}}$ |
| SMERL margin | $0.1 R_{\text{SAC}}$ |

Table E.6: REVERSE SMERL hyperparameters

| Parameter | Value |
|---|---|
| Actor network | $[256, 256, |\mathcal{A}|]$ |
| Critic network | $[256, 256, 1]$ |
| Batch size | 256 |
| Optimizer | Adam |
| Learning rate | $3 \times 10^{-4}$ |
| Replay buffer size | $10^6$ |
| Discount factor $\gamma$ | 0.99 |
| Target smoothing coefficient $\tau$ | 0.005 |
| Skill distribution | Normal distribution |
| Diversity reward scale | 10.0 |
| SMERL target | $R_{\text{SAC}}$ |
| SMERL margin | $0.1 R_{\text{SAC}}$ |

Table E.7: DCG-ME hyperparameters

| Parameter | Value |
|---|---|
| Number of centroids | 1024 |
| Evaluation batch size $b$ | 256 |
| Policy networks | $[128, 128, |\mathcal{A}|]$ |
| Number of GA variations $g$ | 128 |
| GA variation param. 1 $\sigma_1$ | 0.005 |
| GA variation param. 2 $\sigma_2$ | 0.05 |
| Actor network | $[256, 256, |\mathcal{A}|]$ |
| Critic network | $[256, 256, 1]$ |
| TD3 batch size $N$ | 100 |
| Critic training steps $n$ | 3000 |
| PG training steps $m$ | 150 |
| Optimizer | Adam |
| Policy learning rate | $5 \times 10^{-3}$ |
| Actor learning rate | $3 \times 10^{-4}$ |
| Critic learning rate | $3 \times 10^{-4}$ |
| Replay buffer size | $10^6$ |
| Discount factor $\gamma$ | 0.99 |
| Actor delay $\Delta$ | 2 |
| Target update rate | 0.005 |
| Smoothing noise var. $\sigma$ | 0.2 |
| Smoothing noise clip | 0.5 |
| lengthscale $l$ | 0.008 |
| Descriptor sigma $\sigma_d$ | 0.0004 |

Table E.8: UVFA hyperparameters

| Parameter | Value |
|---|---|
| Actor network | $[512, 512, |\mathcal{A}|]$ |
| Critic network | $[512, 512, 1]$ |
| Lagrangian network | $[8, 1]$ |
| Imagination batch size $N$ | 1024 |
| Real environment exploration batch size | 16 |
| Total timesteps | $1 \times 10^7$ |
| Optimizer | Adam |
| Learning rate | $3 \times 10^{-4}$ |
| Replay buffer size | $10^6$ |
| Discount factor $\gamma$ | 0.997 |
| Imagination horizon $H$ | 15 |
| Target smoothing coefficient $\tau$ | 0.005 |
| Sampling period $T$ | 100 |
| Lambda Return $\lambda$ | 0.95 |
| Lagrange multiplier $\lambda_{\mathcal{L}}$ | 0.66 |

Table E.9: SEP-**z** hyperparameters

| Parameter | Value |
|---|---|
| Actor network | $[512, 512, |\mathcal{A}|]$ |
| Critic network | $[512, 512, 1]$ |
| Lagrangian network | $[8, 1]$ |
| Imagination batch size $N$ | 1024 |
| Real environment exploration batch size | 16 |
| Total timesteps | $1 \times 10^7$ |
| Optimizer | Adam |
| Learning rate | $3 \times 10^{-4}$ |
| Replay buffer size | $10^6$ |
| Discount factor $\gamma$ | 0.997 |
| Imagination horizon $H$ | 15 |
| Target smoothing coefficient $\tau$ | 0.005 |
| Sampling period $T$ | 100 |
| Lambda Return $\lambda$ | 0.95 |

Table E.10: FIXED-$\lambda$ hyperparameters

| Parameter | Value |
|---|---|
| Actor network | $[512, 512, |\mathcal{A}|]$ |
| Critic network | $[512, 512, 1]$ |
| Successor Feature network | $[512, 512, |\mathcal{Z}|]$ |
| Lagrangian network | $[8, 1]$ |
| Imagination batch size $N$ | 1024 |
| Real environment exploration batch size | 16 |
| Total timesteps | $1 \times 10^7$ |
| Optimizer | Adam |
| Learning rate | $3 \times 10^{-4}$ |
| Replay buffer size | $10^6$ |
| Discount factor $\gamma$ | 0.997 |
| Imagination horizon $H$ | 15 |
| Target smoothing coefficient $\tau$ | 0.005 |
| Sampling period $T$ | 100 |
| Lambda Return $\lambda$ | 0.95 |
| Lambda $\lambda_{\mathcal{L}}$ | 0.5 |

## F    EVALUATION METRICS DETAILS

In this section, we illustrate how to compute and read the distance and performance profiles in Figure 2. In the QD community, there is a consensus that a convenient evaluation metric for assessing the performance of algorithms is the "distance/performance profile" (Flageat et al., 2022; Grillotti et al., 2023; Grillotti & Cully, 2022a; Batra et al., 2023), and is also being used in skill learning for robotics (Margolis et al., 2022). The profiles are favored because it effectively captures the essence of what QD algorithms aim to achieve: not just finding a single optimal solution but exploring a diverse set of high-quality solutions.

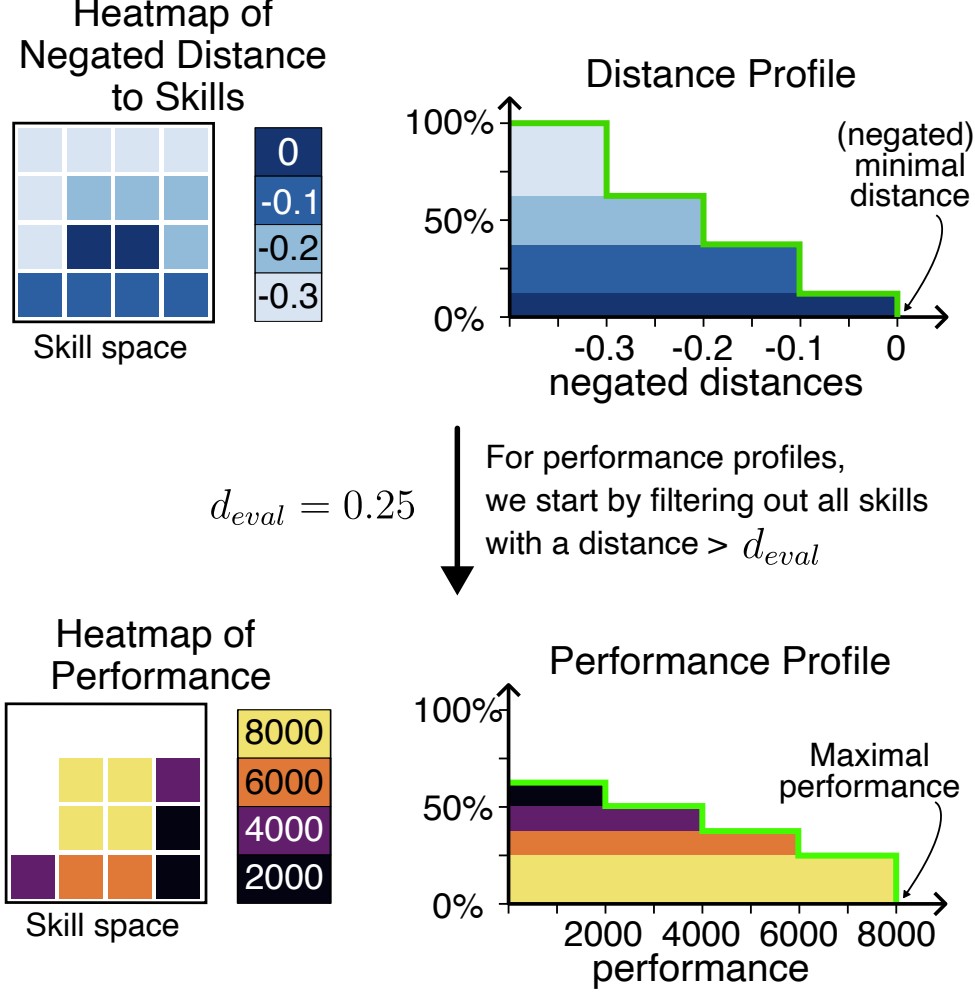

Figure F.22: **Right** *Distance* and *performance* profiles obtained from **Left** Heatmaps of negated distance to skills and performance. A skill $\mathbf{z}$ is considered successfully executed if $d(\mathbf{z}) < d_{\text{eval}}$, otherwise it is considered failed. All the skills that are not successfully executed by the policy are filtered out before computing the performance heatmap and profile. For a given distance $d$, the distance profile shows the proportion of skills with distance lower than $d$. For a given performance $p$, the performance profile shows the proportion of skills with performance higher than $p$. This figure also illustrates how to read the performance of the highest-performing skill (maximal performance of the agent) and the distance to skill of the best executed skill (minimal distance to skill of the agent).

