# OpenReview forum: "Skill-Conditioned Policy Optimization with Successor Features Representations"
_ICLR.cc/2024/Conference — ICLR 2024 Conference Withdrawn Submission_

### Official Review · Reviewer_XD7J · 2023-10-31

**Soundness:** 2 fair
**Presentation:** 3 good
**Contribution:** 2 fair
**Rating:** 5
**Confidence:** 4

**Summary:**

The paper introduces an HRL algorithm for discovering skills based on the successor features. The paper claims to extend the traditional concept of goal to trajectory-based skill. The proposed algorithm relies on constrained optimization to balance maximizing the environment rewards and the skills' intrinsic rewards. The authors present result for continuous control task, showing both qualitative and quantitative results on the Brax physics engine.

**Strengths:**

- The authors rely on constrained optimization to balance the extrinsic rewards and the intrinsic rewards. In practice this is implemented through a Lagrangian relaxation.
- The experiments follow a rigorous protocol using multiple seeds (10)
- The authors provide both quantitative and qualitative results

Usually methods in HRL simply add the extrinsic reward to the intrinsic rewards without considerations to the stability and converge of the method. Constrained optimization offers a more rigorous way to do so. Moreover, the paper shows good scientific standards in the presentation of the empirical results, which has been a weakness for many recent papers in the field.

**Weaknesses:**

- One of the main claims of the paper is that SCOPE "extends the traditional concept of goal to trajectory-based skill." This is simply wrong and clearly misrepresents a large body of literature on goal-conditioned RL and HRL in general
- The authors compare to a few HRL algorithms that learn skills, these algorithms have their rewards defined through DIAYN. This is an outdated baseline that has been beaten numerous times in the literature
- The qualitative results do not convey very effectively the diversity of the learned skills

The claim that the proposed method extends the traditional concept of goal to trajectory level skill seems to me like a fundamental misunderstanding of what goals can represent in HRL. The idea that goals are associated with a particular state (meaning x,y,z positions) is a very narrow point of view of the numerous ways they are instantiated in the literature. Goals essentially depend on the goal space, which can encode any kind information we could imagine, including temporally extended features such as speed or feet contact rate.

The algorithm is compared to a variety of algorithms, but the HRL baselines essentially build their reward on DIAYN, which is not a good algorithm anymore. A recent state-of-the-art algorithm building on a quantity closely connected to the successor features is the idea of Laplacian based options [1]. A comparison to such a strong algorithm would be necessary to really attest to the algorithm's performance.

The qualitative experiments do not convince me that the algorithm learns diverse skills. Figure 2 is very poorly presented, how is the distance exactly calculated? What is the x axis? Why use a task dependent d_eval? It is very unclear how this exactly relates to diversity. Previous work [2] presents diversity in a better way and such visualization would help the paper a lot.

**Questions:**

How are the skills chosen exactly? Do they use the hand-defined features from Section 5.1 (feet contact, angle, ...)? How does the method avoid learning multiple times the same skill?

Why refer to DOMiNO as Reverse SMERL? DOMiNO presents strong results using methods that do not rely on DIAYN, this would be a much better comparison.

More details on Figure 5 would be needed.

"Solving Problem P1 amounts to maximizing the return while executing a desired skill" This is in essence the main goal of this paper [3].

"SF captures the expected future states visited under a given policy" This is not right, it is the expected future feature occupancy, not the state.

"DOMiNO learns policies that are not skill-conditioned, hence the skills are not controllable." This is not right. Conditioning a policy on a N possible goal is the same as to learning N different discrete skills. It is simply a question of architecture.

Despite building skills on the SF, no mention of [4] is made throughout the paper, which is closely related.

====================================================================================

[1] Deep Laplacian-based Options for Temporally-Extended Exploration. Klissarov and Machado, 2023

[2] Eigenoption Discovery through the Deep Successor Representation. Machado et al. 2018

[3] Reward-respecting subtasks for model-based reinforcement learning. Sutton et al. 2023

[4] Lipschitz-constrained Unsupervised Skill Discovery. Park et al. 2022

---

> ### Author Response · Authors · 2023-11-20
> **Response to reviewer 1/3**
>
> First and foremost, we would like to thank you for your thorough review and insightful comments on our work. Your feedback is invaluable and we appreciate the opportunity to clarify and improve upon the points you have raised, especially about the choice of baselines and about the missing references in the related work.
>
> Following your feedback we have updated our claims to reflect better our contribution and improved our discussion of the algorithms we chose as baselines and why other approaches were not included.
> We agree with the reviewer that the pointed references are relevant for this contribution and included them in the paper.
>
>
> >  One of the main claims of the paper is that SCOPE "extends the traditional concept of goal to trajectory-based skill." This is simply wrong and clearly misrepresents a large body of literature on goal-conditioned RL and HRL in general.
> > The claim that the proposed method extends the traditional concept of goal to trajectory level skill seems to me like a fundamental misunderstanding of what goals can represent in HRL. The idea that goals are associated with a particular state (meaning x,y,z positions) is a very narrow point of view of the numerous ways they are instantiated in the literature. Goals essentially depend on the goal space, which can encode any kind information we could imagine, including temporally extended features such as speed or feet contact rate.
>
> Thank you for your insightful comments regarding our claim, we appreciate the opportunity to clarify our stance.
> - We acknowledge that trajectory-based goals exist in the literature, and we certainly did not intend to imply that trajectory-based goals are non-existent in the field.
> - Rather, we wanted to draw an analogy between the **traditional** concept of goal, where goals are associated with specific states [1, 2, 3] and our method.
> - In GCRL, with this **traditional** concept of goal, the objective is to achieve a desired goal $g$, i.e. to reach a state that is in the subset $\\{\left. s \in \mathcal{S} \right\vert \phi(s) = g\\}$. In contrast, our objective is to execute a desired skill $z$, i.e. to sample a trajectory that is in the subset $\\{\left. (s\_t)\_{t \geq 0} \in \mathcal{S}^\mathbb{N} \right\vert \lim\_{T \rightarrow \infty} \frac{1}{T} \sum\_{t=0}^{T-1} \phi\_t = z\\}$ (see Section 3 for a longer discussion). Therefore, SCOPA tries to reach a goal in average over a trajectory rather than at state level.
> - However, we agree that the sentence is misleading, so we have reformulated the claims about trajectory-based goals in the paper and we have also reformulated a paragraph in Section 3. We hope that this addresses your concerns.
>
> [1] Goal-Conditioned Reinforcement Learning: Problems and Solutions. Liu et al. 2022\
> [2] Universal Value Function Approximators. Schaul et al. 2015\
> [3] Hindsight Experience Replay. Andrychowicz et al. 2017
>
>
> > The authors compare to a few HRL algorithms that learn skills, these algorithms have their rewards defined through DIAYN. This is an outdated baseline that has been beaten numerous times in the literature.
> > The algorithm is compared to a variety of algorithms, but the HRL baselines essentially build their reward on DIAYN, which is not a good algorithm anymore. A recent state-of-the-art algorithm building on a quantity closely connected to the successor features is the idea of Laplacian based options [1]. A comparison to such a strong algorithm would be necessary to really attest to the algorithm's performance.
>
> We understand your concerns, but in the present work, we do not directly compare our approach to DIAYN, but to state-of-the-art algorithms that build on top of DIAYN and that are way closer to our work. Thank you for the reference, however, the code for [1] is not available, so we could not compare to this baseline.

---

> ### Author Response · Authors · 2023-11-20
> **Response to reviewer 2/3**
>
> > The qualitative results do not convey very effectively the diversity of the learned skills
> > The qualitative experiments do not convince me that the algorithm learns diverse skills. Figure 2 is very poorly presented, how is the distance exactly calculated? What is the x axis? Why use a task dependent d_eval? It is very unclear how this exactly relates to diversity. Previous work presents diversity in a better way and such visualization would help the paper a lot.
>
> The qualitative results are inspired by many published papers in different communities, such as Quality-Diversity [1, 2] and GCRL [3].
> In our opinion, the best way to visualize the diversity of learned skills is to watch the videos on our website mentioned in the abstract.
>
> We agree that distance and performance profiles may be difficult to interpret at first sight.
> Those are also inspired by previous works in the Quality-Diversity [2, 4] community and in the GCRL community [3].
> They can be summarized as follows: for a score $x$ on the x-axis, the score profile shows the percentage of the skill space where the score is higher than $x$. Hence, the higher the distance profile, the closer the policy gets to the target skills.
> To make it easier to read, we added an illustration in Appendix F that shows how those distance and performance profiles are plotted, and how to interpret them. This new illustration is now referenced in the caption of Figure 2.
>
> We also added the missing labels in Figure 2. Thank you very much for spotting this.
>
> The idea of $d\_{eval}$ is simple. We intend to calculate the performance of our skill-conditioned policy. However, we want to measure the performance of our policy only on the skills that are successfully executed by the policy. For example, consider a robot whose objective is to minimize energy consumption, and where the skills are target velocities of the robot. If the policy is conditioned on the skill [5., 0.] m/s, but the policy does not move, then it achieves great performance (because it minimizes energy consumption). However, the skill [5., 0.] is not successfully executed. In that case, we want to ignore the performance.
>
> In the end, $d\_{eval}$ is just a tool we use for plotting the performance results. It is used to filter out all the skills that the policy does not execute successfully before plotting the performance heatmaps and profiles.
> For each sampled skill $\\boldsymbol{z}$, we perform rollouts  with $\pi(.|., \\boldsymbol{z})$ and measure the distance between $\boldsymbol{z}$ and $\frac{1}{T}\sum\_{t=1}^T$ $\boldsymbol{\phi}\_t$.
> If the distance is higher than $d_{eval}$, then the skill is not considered as "successfully executed", and we don't take its performance into account.
>
> Across all tasks, $d_{eval}$ is chosen equal to $5\delta$; we tested other values from $\delta$ to $8\delta$, and the conclusions drawn by the plots remained the same.
> However for values of $d_{eval}$ below $3\delta$, the results were merely readable when presented in the paper: the performance profiles of most baselines get very close to 0, and their performance heatmaps are nearly empty, as $d_{eval}$ is too small. You can find the values of $d_{eval}$ given in Appendix C Table C.1.
>
> [1] Robots that can adapt like animals, Cully et al. (2015)\
> [2] Benchmarking quality-diversity algorithms on neuroevolution for reinforcement learning, Flageat et al. (2022)\
> [3] Rapid locomotion via reinforcement learning, Margolis et al. (2022)\
> [4] Don't Bet on Luck Alone: Enhancing Behavioral Reproducibility of Quality-Diversity Solutions in Uncertain Domains, Grillotti et al. (2023)
>
>
> > How are the skills chosen exactly? Do they use the hand-defined features from Section 5.1 (feet contact, angle, ...)? How does the method avoid learning multiple times the same skill?
>
> - The feature functions are user-defined for each task, see Section 5.1 and Appendix C for tasks details.
> - The skill executed over a trajectory is defined as the average of the features. Therefore, the skill space corresponds to all the possible average of sequence of features.
> - The target skills are uniformly sampled in the skill space as explained in Algorithm 1 and in Section 4.2.
> - We could imagine better skill sampling strategies but this is beyond the scope of this work.

---

> ### Author Response · Authors · 2023-11-20
> **Response to reviewer 3/3**
>
> > Why refer to DOMiNO as Reverse SMERL? DOMiNO presents strong results using methods that do not rely on DIAYN, this would be a much better comparison.
>
> We agree with the reviewer that DOMiNO and Reverse SMERL are two different algorithms and we did not intend to say otherwise.
> To make this clear in the paper, we have added a footnote the first time we mention Reverse SMERL: "*originally introduced as a baseline of DOMiNO*".
>
> We originally considered including DOMiNO in the baselines. However, DOMiNO is not designed to learn pre-defined skills like DCG-ME or like SMERL/Reverse SMERL can do by using DIAYN+prior. Consequently, for DOMiNO the notion of "distance to skill" does not exist and thus, the policies cannot be evaluated using traditional QD metrics.
>
>
> > "DOMiNO learns policies that are not skill-conditioned, hence the skills are not controllable." This is not right. Conditioning a policy on a N possible goal is the same as to learning N different discrete skills. It is simply a question of architecture.
>
> We agree with the reviewer that this sentence needs to be updated to improve clarity.
> The new sentence is: *Specifically, DOMiNO trains $N$ policies that are different from one another, but these policies are not trained to execute pre-defined skills.*
> As originally stated, DOMiNO is not designed to control skills, neither this can be done without changing the algorithm fundamentals.
>
> Our evaluation metrics expect the policies to take as input the skill $\boldsymbol{z}$, thus all our baselines return policies which (1) can take skills $\boldsymbol{z}$ as input, and (2) are trained to execute those skills. DOMiNO does not verify that property (see answer above).
>
> Indeed, DOMiNO's policies are trained to maximise the distance in the state-action occupancy of other policies.
> Thus, DOMiNO returns $N$ policies exhibiting diverse behaviours; but those policies are not trained to follow any skill.
> Hence, DOMiNO does not solve the same problem as SCOPA and we believe that a comparison would be unfair to DOMiNO.
>
>
> > Solving Problem P1 amounts to maximizing the return while executing a desired skill" This is in essence the main goal of this paper [3].
>
> Thank you for the reference, we have added a citation in the related works section.
>
>
> > More details on Figure 5 would be needed.
>
> We added more details to the caption of Figure 5.
>
>
> > Despite building skills on the SF, no mention of [4] is made throughout the paper, which is closely related.
>
> That is indeed very related even though [4] learns its features, while ours are pre-defined.
> Thank you very much, we have added a reference to [4] in the related work section of the new version.
>
>
> > "SF captures the expected future states visited under a given policy" This is not right, it is the expected future feature occupancy, not the state.
>
> Thank you very much for spotting this mistake. We fixed it in the new version.
>
>
> We once again thank you for your constructive critique. Your feedback has been instrumental in refining our paper, and we hope that our responses adequately address your concerns. We are committed to contributing valuable and clear research to the field, and your input has been essential in guiding our efforts towards this goal. Should you have any remaining doubts or concerns, please do not hesitate to share them with us. We will be more than happy to incorporate your feedback, as we did with the initial review, to further improve the quality of our work.

---

> > ### Comment · Reviewer_XD7J · 2023-11-20
> >
> > Dear Authors,
> >
> > thank you for your detailed rebuttal! We appreciate your updated manuscript and seen some positive improvements.
> >
> > >"extends the traditional concept of goal to trajectory-based skill."
> >
> > I appreciate that the authors are receptive to this criticism. I agree that the traditional idea of goals is tied to states, but this clearly has been generalized many times in the literature (please see [1]). Reading the paper I still see instances where the authors claim that they generalize the idea of goals. I find these claims to be misleading. I would encourage the authors to make a deeper pass on the paper.
> >
> > > Thank you for the reference, however, the code for [1] is not available, so we could not compare to this baseline.
> >
> > A simple google search reveals otherwise as shown by [paperswithcode](https://paperswithcode.com/paper/deep-laplacian-based-options-for-temporally).
> >
> > > Thus, DOMiNO returns  policies exhibiting diverse behaviours; but those policies are not trained to follow any skill.
> >
> > I am not sure I understand what the authors mean. What is the meaning of "following any skill"? Policies do not follow skills, they follow rewards. DOMiNO defines a type diversity through a reward function. SCOPA defines another type of diversity through a different reward function. Either method could be better adapted for particular setting.
> >
> > There are additional claims about DOMiNO that I find incorrect. In particular "the set of policies discovered by DOMINO is finite and the diversity is not interpretable." Are the authors claiming that the policies discovered by SCOPA are all interpretable? Looking at the visualizations on the webpage I would beg to differ. More generally, since SCOPA relies on choosing in advance the features that will define the skills (as shown in Appendix C), it is really hard to make the claim that the proposed reward function is the main reason why diversity might be interpretable. To see my point, do the authors expect interpretable skills to emerge if the features used are all of the input features? If not, then this points to the fact that human prior knowledge is needed to define interpretable diversity - which is not only a fact for SCOPA but any algorithm that seeks diversity.
> >
> > >  To make it easier to read, we added an illustration in Appendix F that shows how those distance and performance profiles are plotted, and how to interpret them. This new illustration is now referenced in the caption of Figure 2.
> >
> > Thank you for the explanation, this has helped understand the results of the paper. I am still wondering why use a threshold that is task specific? This seems like a big limitation if one expects to scale this method or simply use it in a different domains. Couldn't the results be presented without this hyper parameter? I would at least like to see these results in the appendix.
> >
> > Another question I have is how is the distance to skill calculated for baselines that use DIAYN/empowerment?
> >
> > Also, are the skills "z" randomly sampled in the feature space? I would a bit more details.
> >
> > What does 'few-shot' exactly mean? How many samples? What about 'hierarchical learning'? For the latter, have you tried with flat RL baselines?

---

> > > ### Author Response · Authors · 2023-11-21
> > > **Response to reviewer 1/4**
> > >
> > > First and foremost, we would like to thank you for your quick reply and insightful comments on our work.
> > >
> > >
> > > > I appreciate that the authors are receptive to this criticism. I agree that the traditional idea of goals is tied to states, but this clearly has been generalized many times in the literature (please see [1]). Reading the paper I still see instances where the authors claim that they generalize the idea of goals. I find these claims to be misleading. I would encourage the authors to make a deeper pass on the paper.
> > >
> > > We are sorry for the misunderstanding. We were not denying the existing literature about trajectory-based goal. We were simply drawing an analogy between state-based goals and our method which tries to reach a state-based goal but "in average" over the trajectory.
> > >
> > > Thank you for spotting remaining misleading statements in the paper. We have made a second pass and have removed, or reformulated the statements.
> > >
> > >
> > > > A simple google search reveals otherwise as shown by paperswithcode.
> > >
> > > We are sincerely sorry for overlooking that. We only searched for references in the paper, and directly on GitHub, but we could not find any repository with the corresponding name.
> > >
> > > At first, DCEO [1] maximizes intrinsic rewards to learn a finite set of options.
> > > Then, it uses those options to perform temporally-extended exploration, which helps maximizing its extrinsic reward on a given task.
> > > In that sense, the problem DCEO solves is not exactly the same as the ones detailed in our Problem Formulation.
> > >
> > > Conceptually speaking, SCOPA solves an RL task with diverse behaviours/skills; while DCEO leverages reward-agnostic options to be more efficient at solving an RL task.
> > > In this work, we compare to algorithms that solve the same kind of problem as SCOPA (e.g. SMERL and DCG-ME).
> > >
> > > Nonetheless, we completely agree on the fact that DCEO is related to our work, and that it should be cited in our Related Works section.
> > > We added the following sentence at the end:
> > > _Another line of work consists of learning diverse exploring options, which can then be used in a temporally-extended manner to improve the efficiency of RL algorithms (Klissarov & Machado, 2023)_
> > >
> > >
> > > [1] Deep Laplacian-based Options for Temporally-Extended Exploration, Klissarov et al. 2023

---

> > > ### Author Response · Authors · 2023-11-21
> > > **Response to reviewer 2/4**
> > >
> > > > Thank you for the explanation, this has helped understand the results of the paper. I am still wondering why use a threshold that is task specific? This seems like a big limitation if one expects to scale this method or simply use it in a different domains. Couldn't the results be presented without this hyper parameter? I would at least like to see these results in the appendix.
> > >
> > > First, I would like to clarify that there are two different parameters that should not be confused, $\delta$ and $d_\text{eval}$.
> > > - $\delta$ is the equivalent of $\epsilon$ in Equation 6 in [1]. $\epsilon$ is a parameter that is commonly used in GCRL to quantify the maximum acceptable distance between the desired goal and the current achieved goal. Indeed, it is generally very difficult to perfectly reach a goal $g$, so a common approach is to consider that a goal is achieved when $||\phi(s_{t+1}) - g|| \leq \epsilon$. $\delta$ fulfills exactly the same role as $\epsilon$ in our method.
> > > You can now see why the value of $\delta$ (or $\epsilon$) depends on the task. For a task where the skills (or goals) are velocities that range between -5. and 5 m/s, a reasonable threshold would be 0.5 m/s for example. However, for a task where the skills are feet contact rates that range between 0. and 1., a reasonable threshold would be 0.01 for example. Basically, it is up to the user to decide what is a reasonable threshold for his application. However, a general heuristic we used for SCOPA, is to take the side length of the skill space and divide by 100. We are not concerned about any limitation about such parameter as it is very commonly used, not very sensitive and very natural to choose for the user.
> > > - $d_\text{eval}$ is only used for plotting the results and is **not** an hyperparameter of the algorithm. The purpose of $d_\text{eval}$ is simply to filter out the skills that are not successfully executed. Indeed, if we sample a skill $z_\text{desired}$ and perform a rollout with the policy conditioned on the skill z, we will observe a trajectory that achieves a certain return $R$ and a certain skill $z_\text{achieved}$. However, if the achieved skill does not correspond to the desired skill, we consider that the skill is not successfully executed and we ignore the return R.
> > >
> > > The Ant - Velocity example is the perfect task to illustrate why it is necessary to filter out the skills that are not successfully executed. In this task, the objective is to minimize energy consumption, and the skills are the target velocities $v = [v_x, v_y]$. Imagine an Ant whose policy is to do nothing and stay at the origin. Then, the ant is only able to achieve the skill [0., 0.]. However, without filtering it would achieve maximum return for all skills because the optimal policy for the reward is to not move to minimize energy consumption.
> > >
> > > The way to measure performance is in accordance with SCOPA's philosophy and problem formulation P1. We want to learn specific skills (defined by the feature function), but we want to find the highest performing (according to the reward) way to execute those skills. In Problem P1, the term for executing the skill corresponds to the **constraint**, which means that executing the skills is paramount compared to maximizing return. More generally, in constrained optimization, even if a solution is extremely good, it needs to satisfy the constraint to matter, and that is why SCOPA can execute skills that are opposite to maximizing the return as demonstrating in Section 5.4 with Ant - Velocity.
> > >
> > > [1] Goal-Conditioned Reinforcement Learning: Problems and Solutions. Liu et al. 2022\
> > > [2] Uncertain Quality-Diversity: Evaluation methodology and new methods for Quality-Diversity in Uncertain Domains. Flageat et al. 2023\
> > > [3] Don't Bet on Luck Alone: Enhancing Behavioral Reproducibility of Quality-Diversity Solutions in Uncertain Domains. Grillotti et al. 2023\
> > > [4] Rapid locomotion via reinforcement learning, Margolis et al. (2022)

---

> > > ### Author Response · Authors · 2023-11-21
> > > **Response to reviewer 3/4**
> > >
> > > > Thank you for the explanation, this has helped understand the results of the paper. I am still wondering why use a threshold that is task specific? This seems like a big limitation if one expects to scale this method or simply use it in a different domains. Couldn't the results be presented without this hyper parameter? I would at least like to see these results in the appendix.
> > >
> > > The following interpretation of SCOPA's contribution might clarify further:
> > > - In RL, taking the action that brings the most reward, at each step, is a suboptimal policy and thus, cannot be solved with a greedy strategy. The purpose of value functions is to transform a non-greedy problem into a greedy one. Indeed, if the value function of the optimal policy is known, we can use a greedy strategy to get the optimal policy.
> > > - In GCRL, taking the action that decreases the most the distance with the goal, at each time step, is a suboptimal policy to reach the goal. This strategy corresponds to Equation 7 in [1] and is equivalent to maximizing a reward that corresponds to the negated distance between the desired goal and the current achieved goal $-d(\phi(s_{t+1}), g)$. This stategy cannot solve tasks where it first needs to increase the distance to the goal before finally reaching it.
> > > - To overcome this issue, we derive a policy skill improvement update based on successor features that is analogous to the classic policy improvement update. The same way that value functions transform maximizing the return into a greedy strategy, we use successor features to transform executing a skill (i.e. achieving a goal) into a greedy strategy.
> > >
> > > One of the strengths of SCOPA is to be able to execute skills accurately and consistently by leveraging a novel "policy skill improvement" update analogous to traditional policy improvement. Naturally, we want to measure skill execution, which is evaluated with our "distance to skill" metric. In Quality-Diversity optimization, making policies more reproducible in terms of skill execution is an open question [2, 3] and the distance to skill metric is a common metric [2, 3, 4].
> > >
> > > **Having said that, we also think that the results without $d_\text{eval}$ are interesting, and we have included them in Appendix A.3 in the new submission, as requested.**
> > >
> > > [1] Goal-Conditioned Reinforcement Learning: Problems and Solutions. Liu et al. 2022\
> > > [2] Uncertain Quality-Diversity: Evaluation methodology and new methods for Quality-Diversity in Uncertain Domains. Flageat et al. 2023\
> > > [3] Don't Bet on Luck Alone: Enhancing Behavioral Reproducibility of Quality-Diversity Solutions in Uncertain Domains. Grillotti et al. 2023\
> > > [4] Rapid locomotion via reinforcement learning, Margolis et al. (2022)

---

> > > ### Author Response · Authors · 2023-11-21
> > > **Response to reviewer 4/4**
> > >
> > > > Another question I have is how is the distance to skill calculated for baselines that use DIAYN/empowerment?
> > >
> > > This is an excellent question because it is less obvious to calculate the distance to skill for SMERL/Reverse SMERL than it is for DCG-ME or SCOPA. Although, there is a rigorous way to do it. We answer that question very thoroughly in Appendix D.1, we really recommend that you check that section but we also give a shorter answer here.
> > >
> > > In SMERL and Reverse SMERL:
> > > - the actor is conditioned on skills $z_\text{DIAYN}$, from an unsupervised skill space defined by the discriminator $q(z_\text{DIAYN} | s)$, and that is different from SCOPA's skill space.
> > > - the discriminator $q(z_\text{DIAYN} | s)$ (or $q(z_\text{DIAYN} | \phi(s))$ when using prior) gives the distribution of skills $z_\text{DIAYN}$ given $\phi(s)$.
> > >
> > > Therefore, the discriminator is a bijection between the two skill spaces $z_\text{DIAYN}$ and $z$. To execute a particular skill associated with features $z = \phi$, we can simply find the corresponding skill $z_\text{DIAYN}$ with maximum likelihood for the distribution $q(z_\text{DIAYN} | z)$. Finally, we can rollout the policy with that skill $\pi(. | ., z_\text{DIAYN})$ and calculate the distance to skill.
> > >
> > >
> > > > Also, are the skills "z" randomly sampled in the feature space? I would a bit more details.
> > >
> > > The skills are randomly uniformly sampled in the skill space, which is defined as the convex hull of the feature space, as each skill is an average of features (see the end of Section 3 Problem Formulation).
> > > In the paper, we use the notation $z \sim \mathcal{U}(\mathcal{Z})$.
> > > You can see exactly when skills are sampled in Algorithm 1 Line 3 and Line 9.
> > > We also detail the algorithm in Section 4.2 Training.
> > >
> > > Please let us know if you need more information about the skill sampling strategy we use in SCOPA.
> > >
> > >
> > > > What does 'few-shot' exactly mean? How many samples? What about 'hierarchical learning'? For the latter, have you tried with flat RL baselines?
> > >
> > > Few-shot adaptation refers to the ability of a policy to adapt to a new environment without re-training, only evaluation of skills are allowed. Specifically, for each replication (10 random seeds) of each algorithm, we evaluate 2,500 skills uniformly sampled from the skill space and select the performance of the best performing skill.
> > >
> > > For hierarchical learning, for each replication (10 random seeds) of each algorithm, we take the pre-trained policy and we use it to create a wrapper around the environment. Thus, the new environment is the same except that the action space is changed to the skill space. Then, we simply train SAC on that environment and report the average return over 256 rollouts, at the end of training.
> > >
> > > No, we have not compared to a flat RL baseline, because what we want to evaluate is the quality of the discovered skills for each algorithm. We do not claim that the task Ant - Obstacle is impossible to solve for a flat RL algorithm. Rather, we want to show that the skills learned with SCOPA are better to adapt to new environment. This supports the claim that SCOPA can accurately and consistently execute learned skills and that the skills are executed in a way to maximize return at the same time (with constrained optimization).
> > >
> > >
> > > We once again thank you for your constructive critique. Your feedback has been instrumental in refining our paper, and we hope that our responses adequately address your concerns.

---

> > > ### Author Response · Authors · 2023-11-23
> > > **Response to reviewer**
> > >
> > > Thank you for your comments about DOMiNO, we appreciate the opportunity to clarify our stance.
> > >
> > > > I am not sure I understand what the authors mean. What is the meaning of "following any skill"? Policies do not follow skills, they follow rewards. DOMiNO defines a type diversity through a reward function. SCOPA defines another type of diversity through a different reward function. Either method could be better adapted for particular setting.
> > >
> > > DOMiNO learns a set of diverse and near-optimal policies, but the policies discovered are not trained to execute specific target skills. Thus, the method differs from traditional QD methods. Consequently, for DOMiNO the notion of "distance to skill" does not exist and thus, the policies cannot be evaluated using traditional QD metrics.
> > >
> > >
> > > > There are additional claims about DOMiNO that I find incorrect. In particular "the set of policies discovered by DOMINO is finite and the diversity is not interpretable." Are the authors claiming that the policies discovered by SCOPA are all interpretable? Looking at the visualizations on the webpage I would beg to differ. More generally, since SCOPA relies on choosing in advance the features that will define the skills (as shown in Appendix C), it is really hard to make the claim that the proposed reward function is the main reason why diversity might be interpretable. To see my point, do the authors expect interpretable skills to emerge if the features used are all of the input features? If not, then this points to the fact that human prior knowledge is needed to define interpretable diversity - which is not only a fact for SCOPA but any algorithm that seeks diversity.
> > >
> > > We apologize for the misunderstanding, in this context the term "interpretable" is confusing. By saying "the set of policies discovered by DOMINO is finite and the diversity is not interpretable.", we do **not** mean that the policies discovered by SCOPA are all interpretable.
> > >
> > > With that claim, the idea we are trying to convey is the same as the point made above: DOMiNO learns a set of diverse and near-optimal policies, but the policies discovered are not trained to execute **specific** target skills. Indeed, the behaviors of the policies stem from the diversity optimization process and is not given explicitly.
> > >
> > > For example, in the case of Ant Velocity, SCOPA learns to execute specific skills like $z = [5 \text{m/s}, \text{0 m/s}]$. Therefore, when we roll out the policy in the environment with a specific skill $z$, the skill is "interpretable", we know what it means and what to expect from the policy.
> > >
> > > However, for DOMiNO, we don't know beforehand what will be the skill executed by the policy as it results from the diversity optimization process. For that reason, we called the skills of DOMiNO "not interpretable", which was very confusing.
> > >
> > > We have changed the corresponding paragraph with a better formulation that now reads as follows: "...but the policies discovered are not trained to execute specific target skills."

---

### Official Review · Reviewer_a9mP · 2023-10-31

**Soundness:** 2 fair
**Presentation:** 2 fair
**Contribution:** 3 good
**Rating:** 5
**Confidence:** 4

**Summary:**

The paper proposes a novel method for simultaneously learning to solve a given task while also learning a diverse set of skills. Precisely, given the relevant features per state, the paper defines skills as the average of the set of features encountered by a policy. They then define a constrained optimisation problem where for each skill, an agent needs to find a policy that maximises the reward returns and whose average trajectory features converge to the given skill. They then solve a Lagrangian relaxation of the problem by combining several methods from prior works. They use universal value function approximations (UVFAs) to learn the skill-conditioned policies, universal successor features approximations (USFAs) to learn the skill-conditioned successor features, and model-based RL (DreamerV3) to improve sample efficiency. They theoretically show that a solution to the formulated problem exists under some assumptions, and empirically show that their practical algorithm (SCOPA) outperforms prior works in various continuous control tasks.

**Strengths:**

- The paper is mostly well-written and investigates an important problem. The specific proposed approach for learning to solve a given task while learning diverse skills is novel and interesting.

- The authors provide theory showing that under strong assumptions about the environment (mainly the existence of a policy that satisfies hypotheses 1 and 2 simultaneously), their formulated constrained optimisation problem is solvable.

- I really liked the figures visualising the number and quality of skills successfully learned (like fig 3). They are a nice way of visualising the diversity and quality of learned skills. They also show that SCOPA outperforms the baselines in both diversity and quality.

- The paper shows numerous empirical results comparing SCOPA with many relevant baselines in several continuous control tasks. Impressively, SCOPA is shown to outperform all these baselines in terms of the number/quality of skills learned and usefulness for downstream tasks.

**Weaknesses:**

**MAJOR**

1- **Theory**:

- It is not clear if the MDPs are assumed to be continuous, discrete, or finite for the theoretical results. This should be made explicit.

- The motivation for the given definition of a skill is not given. It is not clear why it is an average instead of a sum or any other function over the sequence of features. It is also not clear why the average gives the same probability to each feature in the trajectory. Shouldn't it be an expectation over the sum of features?

- The explanations for Proposition 1 and Algorithm 1 are scattered. This made hypothesis 1 of Proposition 1 especially hard to understand.

2- **Baselines and empirical significance**:

  - It is not clear if the 10 replications/runs used for each plot are different runs of each algorithm (e.g Algorithm 1 for SCOPA), or if they are using the same models trained from a single run. This should be clarified, as it affects the significance of the empirical results.

  - The authors use MBRL (DreamerV3) to speed up the learning of their actors and critics, but do not do the same for the main baselines (SMERL, REVERSE SMERL, and DCG-ME). This makes the experiments extremely unfair towards the baselines. If the authors did not want to go through the additional effort of augmenting the baselines with DreamerV3, then I think they also shouldn't have augmented SCOPA with DreamerV3.

  - The authors used actor/critic networks with hidden layers of size 512 for SCOPA (and its variants), but only 256 for the main baselines (SMERL, REVERSE SMERL, and DCG-ME). The authors did not explain why they made this decision. Additionally, the number of iterations used for the main baselines (or possibly function evaluations for DCG-ME) was not stated.

  - These make it hard to evaluate the significance of the empirical results, and I think it also puts into question the claims and conclusions based on them.

3- **Evaluation Metrics**:

  - The *distance to skill* and *performance* metrics used are specific to the type of skills learned by SCOPA. Despite this, the authors spend a lot of time comparing their approach to the baselines using these metrics, which is extremely unfair to the baselines. Naturally, this made the baselines look extremely terrible, inconsistent with the great results shown in those prior works.

  - Just like prior works, I think the authors should have focused more on the "Using skills for adaptation" experiments for the comparison with the baselines. These experiments demonstrate how useful the diverse skills learned by each method are for downstream tasks. The authors should include such experiments for a variety of downstream tasks (similar to prior works).

  - While the authors have attempted to make their evaluation metrics fair by modifying the main baselines (SMERL, REVERSE SMERL, and DCG-ME) to learn the same skills that SCOPA is trying to learn, I think this is extremely wrong. While giving the baselines the same additional priors as SCOPA is good, the authors shouldn't have modified the actual optimisation objectives or skills being learned them.


**MINOR**

- The proposed approach relies on the state features being predefined/known.
- The abstract and introduction could do a better job of explaining the contributions and what SCOPA is since I couldn't tell the difference between it and UVFAs or USFAs until Sec 3.
- Some of the figures in the experiments section are too far from where they are referenced for the first time. E.g Fig 2 and 3.
- There are a couple of typos or grammar errors throughout the paper. I have listed some examples here
  - Page 6: There is no fullstop after the definitions of $r_{SMERL}$ and $r_{REVERSE}$.
  - Page 5 "critic training": There is a "where" or "and" missing between the two equations.

**Questions:**

It would be great if the authors could address the major weaknesses I outlined above. I am happy to increase my score if they are properly addressed, as I may have misunderstood pieces of paper.

---

> ### Author Response · Authors · 2023-11-20
> **Response to Reviewer a9mP (Part 1/4)**
>
> First and foremost, we would like to thank you for your thorough review and insightful comments on our work. Your feedback is invaluable and we appreciate the opportunity to clarify and improve upon the points you have raised.
>
> **MAJOR**
>
> 1- **Theory**:
>
> > *It is not clear if the MDPs are assumed to be continuous, discrete, or finite for the theoretical results. This should be made explicit.*
>
> In the original proposition you reviewed, the MDP was considered continuous. However, to improve the clarity of section 4.1, we have replaced the proposition and moved the original one to Appendix B.2.
>
> The new proposition is easier to understand and assumes the MDP to be discrete; and this is now explicit.
>
>
> > *The motivation for the given definition of a skill is not given. It is not clear why it is an average instead of a sum or any other function over the sequence of features. It is also not clear why the average gives the same probability to each feature in the trajectory. Shouldn't it be an expectation over the sum of features?*
>
> We agree that the motivation for the definition of skill in our original submission was lacking. We have updated Section 3 with better notations, two examples of tasks with two different types of skill and an analogy between the definition of goal found in GCRL literature and our definition of skill to help the readers understand better.
>
> We define the skill executed by the policy as the expected features under the policy's stationary distribution. The expected features have been used before in [1, 2, 3] to characterize the behavior of the policy. However, to the best of our knowledge, we are the first to condition the policy on the expected features.
>
> In particular, the reason why we do not want to take the expectation over the sum of features, is because
> -  In the non-discounted case, the infinite sum of features may diverge.
> -  In the discounted case, the infinite sum is not as easily-interpretable as the average. For example, in the first example given in Section 3 where the features characterize the velocity of a robot, it makes sense to try to follow a velocity in average, but not in sum.
>
> Thank you for pointing that the explanation was unclear. Please, let us know if you have any remaining doubts or concerns.
>
> [1] Improving Generalization for Temporal Difference Learning: The Successor Representation. Dayan. 1993\
> [2] Successor Features for Transfer in Reinforcement Learning. Barreto et al. 2017\
> [3] Discovering Policies with DOMiNO: Diversity Optimization Maintaining Near Optimality. Zahavy et al. 2022
>
>
> > *The explanations for Proposition 1 and Algorithm 1 are scattered. This made hypothesis 1 of Proposition 1 especially hard to understand.*
>
> We agree that the way the proposition and the algorithm were introduced was a bit unclear.
> We have updated Section 4.1 to make the proposition more straightforward and now reference Algorithm 1 in section 4.2.
> We hope that the paper have gained in clarity and readability, please let us know if you have any remaining concerns.
>
>
> 2- **Baselines and empirical significance**:
>
> > *It is not clear if the 10 replications/runs used for each plot are different runs of each algorithm (e.g Algorithm 1 for SCOPA), or if they are using the same models trained from a single run. This should be clarified, as it affects the significance of the empirical results.*
>
> Each experiment is replicated 10 times with independent random seeds, and each replication learns a different model from scratch.
> We have now added the precision: *"Each experiment is replicated 10 times with random seeds and each replication is independent and trains all components from scratch."* at the beginning of Section 5.3 Evaluation Metrics.
>
> To evaluate the distance to skill metrics and performance metrics we perform 10 rollouts for each algorithms and each skills, for each of the 10 replications.

---

> ### Author Response · Authors · 2023-11-20
> **Response to Reviewer a9mP (Part 2/4)**
>
> > *The authors use MBRL (DreamerV3) to speed up the learning of their actors and critics, but do not do the same for the main baselines (SMERL, REVERSE SMERL, and DCG-ME). This makes the experiments extremely unfair towards the baselines. If the authors did not want to go through the additional effort of augmenting the baselines with DreamerV3, then I think they also shouldn't have augmented SCOPA with DreamerV3.*
>
> The world model we use is an integral and essential part of our algorithm, and showing that we can extend DreamerV3 to also predict features and learn a successor features (i.e. a skill critic analogous to a classic performance critic) is one of the main contributions of this paper, which to the best of our knowledge was never done in the litterature.
>
> Furthermore, we trained each baseline until convergence for a maximum budget of 10 million environment interations (which is well enough time for all baselines to converge) and we do not compare the convergence speed of SCOPA against the baselines with learning curves in Section 5.4 Results. Our evaluation methods only focus on the final performance metrics achieved by each algorithm after the completion of the training process.
>
> Finally, we did not want to modify the original implementations of the baselines we compare to, because augmenting the baselines with DreamerV3 runs into implementation issues that are subject to debate on how to actually make it work, and would result in different algorithms entirely.
> For example, DCG-MAP-Elites cannot possibly be extended because it would require performing full rollouts in imagination (see \[1, 2\] for examples of model-based QD algorithm), whereas DreamerV3 performs only 15 steps in imagination and uses bootstrapping to estimate the critics.
>
> [1] Dynamics-Aware Quality-Diversity for Efficient Learning of Skill Repertoires, Lim et al. (2021)\
> [2] Efficient Exploration using Model-Based Quality-Diversity with Gradients, Lim et al. (2023)
>
>
> > *The authors used actor/critic networks with hidden layers of size 512 for SCOPA (and its variants), but only 256 for the main baselines (SMERL, REVERSE SMERL, and DCG-ME). The authors did not explain why they made this decision. Additionally, the number of iterations used for the main baselines (or possibly function evaluations for DCG-ME) was not stated. These make it hard to evaluate the significance of the empirical results, and I think it also puts into question the claims and conclusions based on them.*
>
> SMERL, Reverse SMERL and DCG-ME are all baselines that have been fine-tuned on similar locomotion tasks like Walker, HalfCheetah, Ant, Humanoid. Therefore, we initially made the decision to keep the hyperparameters coming from the original papers for all baselines.
>
> However, we understand your concern and we believe that you raise a very good point. We have run all baselines with hidden layers of [512, 512] and observed a significant decrease in performance for Reverse SMERL and DCG-ME. For SMERL, the performance are not significantly better or worse.
> For this reason, we decided to keep the original architectures for our baselines, and the justification has been added to the hyperparameters section in Appendix.
>
> We think that this additional evaluation increases the significance of the empirical results, and hope that this addresses your concerns.
>
> Furthermore, we now specify the number of environment interactions for all baselines in Appendix E.
> Each algorithm is run for 10 million environment steps, and we ensured that is enough for all of them to converge.

---

> ### Author Response · Authors · 2023-11-20
> **Response to Reviewer a9mP (Part 3/4)**
>
> 3- **Evaluation Metrics**:
>
> > *The _distance to skill_  and  _performance_  metrics used are specific to the type of skills learned by SCOPA. Despite this, the authors spend a lot of time comparing their approach to the baselines using these metrics, which is extremely unfair to the baselines. Naturally, this made the baselines look extremely terrible, inconsistent with the great results shown in those prior works.*
>
> The metrics used in this paper are not new, but instead well established in the Quality-Diversity literature.
> - DCG-ME relies on a descriptor function that must be defined by the user, and that takes a trajectory as input and gives the skill executed over that trajectory as output. Naturally, in the present work, we chose the descriptor function to be the average of the features over the trajectory. Thus, the distance to skill and performance metrics are completely adapted to this baseline. Morever, DCG-ME has been evaluated with similar metrics in its own paper [4].
> - For SMERL and Reverse SMERL, we agree that the metrics are less adapted because they are not per se QD algorithms, but rather algorithms built for robustness using diversity mechanism (namely DIAYN). However, last year a spotlight ICLR paper used rigorously the same metrics to compare SMERL to QD algorithms [5]. However, regardless of QD metrics, we evaluate on downstream tasks too for a full and fair comparison.
>
> To evaluate the skill reachability, we take inspiration from [1], whose graphs are similar to the distance profiles and distance heatmaps used in our paper.
> To evaluate performance across the skill space, the metrics from Quality-Diversity are adapted [2] and have been used in previous work [3].
>
> We understand your concern and believe that you raise a good point. However, that is why we have also evaluated the different approaches on downstream adaptation tasks. Moreover, we would like to highlight that the adaptation tasks are literally borrowed from SMERL paper itself (see Figure 1 for Ant - Obstacle, see Figure 2 for Humanoid - Hurdles and Humanoid - Motor Failure in SMERL paper).
>
> [1] Rapid locomotion via reinforcement learning, Margolis et al. (2022)\
> [2] Benchmarking quality-diversity algorithms on neuroevolution for reinforcement learning, Flageat et al. (2022)\
> [3] Don't Bet on Luck Alone: Enhancing Behavioral Reproducibility of Quality-Diversity Solutions in Uncertain Domains, Grillotti et al. (2023)\
> [4] MAP-Elites with Descriptor-Conditioned Gradients and Archive Distillation into a Single Policy, Faldor et al. (2023)\
> [5] Neuroevolution is a Competitive Alternative to Reinforcement Learning for Skill Discovery, Chalumeau et al. (2023)
>
>
> > *Just like prior works, I think the authors should have focused more on the "Using skills for adaptation" experiments for the comparison with the baselines. These experiments demonstrate how useful the diverse skills learned by each method are for downstream tasks. The authors should include such experiments for a variety of downstream tasks (similar to prior works).*
>
> In our original submission, we had 3 downstream tasks including 1 hierarchical task and 2 few-shot adaptation tasks.
> Since then, we added 2 extra few-shot adaptation tasks:
> - Ant - Gravity task scaling gravity from 0.5 (low gravity) to 3. (high gravity).
> - Walker - Friction task varying the friction with the ground from 0. (no friction) to 5. (high friction).
>
> Hence, our number of downstream tasks is similar to prior work:
> - SCOPA is evaluated on 1 hierarchical task, and 4 few-shot adaptation tasks (**5** different types of perturbation).
> - SMERL has 0 hierarchical tasks, and 9 few-shot adaptation tasks in total (3 different environments times **3** different types of perturbations).
> - [1] compares SMERL and QD algorithms on 1 hierarchical task and 7 few-shot adaptation tasks (**3** kinds of perturbations).
>
> [1] Neuroevolution is a Competitive Alternative to Reinforcement Learning for Skill Discovery, Chalumeau et al. (2023)

---

> ### Author Response · Authors · 2023-11-20
> **Response to Reviewer a9mP (Part 4/4)**
>
> > *While the authors have attempted to make their evaluation metrics fair by modifying the main baselines (SMERL, REVERSE SMERL, and DCG-ME) to learn the same skills that SCOPA is trying to learn, I think this is extremely wrong. While giving the baselines the same additional priors as SCOPA is good, the authors shouldn't have modified the actual optimisation objectives or skills being learned them.*
>
> We are sorry for the misunderstanding but we did **not** modify any of the baselines. All baselines are strictly the same as in their original paper.
>
> We have updated the table D.3 in Appendix with the objective for all baselines. Additionally, you can find more information about the baselines in Section D in Appendix. Especially, a detailed explanation about SMERL's optimization objective is now provided in Section D.1 in Appendix.
>
> If you could specify the section of our paper that caused this confusion, we would be eager to offer a detailed clarification.
>
> **MINOR**
>
> > The proposed approach relies on the state features being predefined/known.
>
> The feature functions are not learned and this is not the focus of the present work, but augmenting SCOPA with unsupervised features is a possible future direction of research, as mentioned in the conclusion.
> Whether it is a limitation or not depends on the need of the user of the algorithm. If the goal of the user is to learn skills like jumping, then it is very useful to be able to provide the feature function directly. However, if the goal of the user is simply to learn skills to adapt to unforeseen situations like changes in friction, then it could be desirable to learn the feature function.
>
>
> > The abstract and introduction could do a better job of explaining the contributions and what SCOPA is since I couldn't tell the difference between it and UVFAs or USFAs until Sec 3.
>
> The abstract and introduction have been reformulated to better emphasize the contributions of SCOPA.
> Thank you for helping clarifying the paper.
>
>
> > Some of the figures in the experiments section are too far from where they are referenced for the first time. E.g Fig 2 and 3.
>
> We have fixed this issue in our updated version.
> Thank you for making the paper better.
>
>
> > There are a couple of typos or grammar errors throughout the paper.
>
> Thank you very much for spotting those typos, they have been corrected.
>
>
> We once again thank you for your constructive critique. Your feedback has been instrumental in refining our paper, and we hope that our responses adequately address your concerns. We are committed to contributing valuable and clear research to the field, and your input has been essential in guiding our efforts towards this goal. Should you have any remaining doubts or concerns, please do not hesitate to share them with us. We will be more than happy to incorporate your feedback, as we did with the initial review, to further improve the quality of our work.

---

### Official Review · Reviewer_9NVL · 2023-11-01

**Soundness:** 4 excellent
**Presentation:** 4 excellent
**Contribution:** 4 excellent
**Rating:** 8
**Confidence:** 4

**Summary:**

This work presents SCOPA, an algorithm to solve tasks given by the task reward while learning skills based on successor features. In this work, a skill is represented as a mean vector of the features (for states or cumulants). The objective (P1) is to learn a skill-conditioned policy to maximize the task return w.r.t such that the mean of the future-occupant features becomes z, which is reformulated as the constraint optimization problem (P2). The authors show that this constraint can be modeled as learning successor features.

**Strengths:**

- The problem is well formulated, and is of a very high significance: This work generalizes and extends goal-conditioned learning in combination with skill learning, and unifies the idea with a very novel, clear and efficient method. Unlike many prior works in (unsupervised) skill learning, this work considers maximization of task rewards and skill learning simultaneously which can result in learning useful skills for fulfilling tasks, which is often hard in unsupervised GCRL.


- The method is very principled and beautifully designed without much hack or ad-hoc techniques. I find the use of SF (successor features) very interesting, and the flow of derivation gave me a lot of joy to read and learn about it. I haven't gone through the details of proof very thoroughly, but at a high level the idea of proof on the Proposition (Problem P1 -> P2) makes sense.


- The algorithm also features intriguing and useful merits and controllability, including diversity and quality trade-off using a single hyperparameter.


- SCOPA works very well on challenging locomotion tasks (including Ant and Humanoid) and their adaptation tasks (e.g. with hurdles, obstacles, and broken body) — prior methods usually have difficulties in learning meaningful skills in these domains. The qualitative results also look impressive.


- The analyses and experiments provided in the paper are very comprehensive and of high quality.

**Weaknesses:**

- The skill space is a linear combination (span) of the feature space, i.e., a skill vector can be represented as only a linear combination of state observations. While this might be limiting in terms of how expressive or arbitrarily complex skill representation can be learned, it's not a major concern because the assumption is quite mild and also a crucial part to make the use of SF possible; in fact, rich skill spaces would be actually more difficult to learn than having such inductive biases.


- To define a skill, so-called "feature engineering" is still needed. In practice, the feature seems to be designed carefully as a low-dimensional subspace of the observation (as low as 2~3 dimensional) in order to be useful. As noted in many unsupervised skill learning papers (namely, Gu et al., 2021, etc.), the method would not work very well if no domain knowledge about which feature spaces would be useful (e.g. velocity for locomotion tasks). This would be a minor limitation, but often in the robotics domain it can be seen as a reasonable assumption.


- While the learning of SF is in principle model-free (although more technically speaking, it's somewhere between model-free and model-based learning), the method requires learning of a world model.

- Figure 2: the x-axis of the plots lacks labels, difficult to figure out what they would exactly mean.
- Proposition: it'd be good to mention what \Psi(st, z) denotes here -- the successor feature.

**Questions:**

Sample efficiency -- how many samples is needed to learn successful skills and solve the task?  Would the method still work if the entire algorithm is trained without the use of model, in a model-free fashion?

---

> ### Author Response · Authors · 2023-11-20
> **Response to Reviewer 9NVL**
>
> First and foremost, we would like to thank you for your thorough review and insightful comments on our work. Your feedback is invaluable and we appreciate the opportunity to improve upon the points you have raised, especially the limitations of the skill definition and the overall clarity of the paper.
>
> > The skill space is a linear combination (span) of the feature space, i.e., a skill vector can be represented as only a linear combination of state observations. While this might be limiting in terms of how expressive or arbitrarily complex skill representation can be learned, it's not a major concern because the assumption is quite mild and also a crucial part to make the use of SF possible; in fact, rich skill spaces would be actually more difficult to learn than having such inductive biases.
>
> If you are interested, we have updated the notations and the comparison between our definition of skill and goals in GCRL in Section 3.
>
>
> > Figure 2: the x-axis of the plots lacks labels, difficult to figure out what they would exactly mean.
>
> Thank you for pointing out inaccuracies in Figure 2, we have added the previously missing x-axis label.
>
>
> > Proposition: it'd be good to mention what \Psi(st, z) denotes here -- the successor feature.
>
> Thank you for your suggestion regarding the clarity of our proposition. We have now clarified the notation Psi in the proposition.
>
>
> > Sample efficiency -- how many samples is needed to learn successful skills and solve the task? Would the method still work if the entire algorithm is trained without the use of model, in a model-free fashion?
>
> - We now specify the number of environment interactions for all baselines in Appendix E. In all experiments, each algorithm is run for a maximum budget of 10 million environment interactions, and we ensured that is enough for all of them to converge.
> - Empirically, we have found out that around 100,000 to 200,000 environment interactions are needed to learn successful skills and to consider the task solved with SCOPA.
> - Currently, the world model is an essential part of our algorithm as SCOPA learns to predict the features in order to estimate the successfor features and be able to backpropagate through the world model with Equation 2.
>
>
> We once again thank you for your constructive critique. Your feedback has been instrumental in refining our paper, and we hope that our responses adequately address your concerns. We are committed to contributing valuable and clear research to the field, and your input has been essential in guiding our efforts towards this goal. Should you have any remaining doubts or concerns, please do not hesitate to share them with us. We will be more than happy to incorporate your feedback, as we did with the initial review, to further improve the quality of our work.

---

### Official Review · Reviewer_SA5X · 2023-11-01

**Soundness:** 2 fair
**Presentation:** 2 fair
**Contribution:** 2 fair
**Rating:** 5
**Confidence:** 3

**Summary:**

This paper introduces Skill-Conditioned OPtimal Agent (SCOPA), which is a method for learning policies that are conditioned on also executing a skill using successor features. Skills are defined as achieving a mean feature vector across a trajectory. SCOPA takes the constrained optimization problem of maximizing reward while achieving a skill and casts it as an optimization problem with a Lagrange multiplier parameter lambda_L that trades off between performance and diversity.

To train policies that skill-conditioned (recall that skills are mean skills), which can be quite costly given the large space of state-skill pairs, a world model is trained using an extension of Dreamer-v3 which can generate training data for the actor-critic training.

The experiments compare SCOPA to related baselines in Walker, Ant, and Humanoid (using Google Brax). The experimental findings suggest that SCOPA is able to achieve skills (including those contrary to the main task), achieves better performance than the baselines, and are more adaptive to downstream tasks.

**Strengths:**

They use multiple baselines for the experiments.

Given their definition of skill, the introduced method seems like a reasonable approach to tackle the problem of achieving reward while learning skills.

Their method does seem to do better than the baselines and achieve desirable characteristics of a skill-learner (good performance, ability to learn skills, adaptability, etc.)

**Weaknesses:**

There are several concerns I have with the paper. In particular, the lack of addressing the limitations of this paper is making me confused. I hope to gain some clarity on this.

Concern - Definition of skill: I don't know if the authors introduced this notion of a skill in this paper, or if there is a precedent in the literature for this definition of skill. If the latter, it would be helpful to include a citation. If the former, I think there needs to be more explanation or justification as this is not intuitive to me and seems to be lacking in several ways. It seems the definition of skill is quite narrow, it defines skills as feature accumulation over an entire trajectory, which does not seem like an intuitive way to capture the notion of a skill in a general sense, and seems useful for a specific type of task (e.g., locomotion tasks as tested in this paper). For example, if the agent must move between points, and the climb a ladder, it already becomes unclear how this impacts how one should define skills.

Concern - Reproducibility: Hyperparameter details are offered in the appendix, which may be enough to reproduce the work. There are indeed many moving parts, such as the actor/critic/world model/SF network, and plenty of hyperparameters that suggest this may not be easy to reproduce. Any thoughts on the authors' behalf regarding reproducibility would be appreciated.

The method seems to hinge on the availability of feature functions to extract relevant limitation. Not much time is spent discussing where this might come from. Is this a limitation? Can these feature functions be learned?

Breadth of experiments: All of the experimental tasks are locomotion related, where there is some task uniformity (i.e., the agent is moving across some plane) that is aligned with the definition of skills. For example, if the agent's task was to walk and then climb a ladder, what skills would be defined/be appropriate? How would SCOPA perform?

I like aspects of this paper, and my concerns may be from confusion or missing the point. However, as I have many questions, I am concerned about the clarity of the paper. If my questions and concerns can be addressed or clarified I may well change my score.

Minor nits:
typo: "dependant" -> "dependent" in figure 2.
typo: "the agent needs multiple timestep" -> "timesteps"
typo: missing a period before "For a fair comparison..."
typo: "extrinsinc" -> "extrinsic"

**Questions:**

1. Will the code be released? While the hyperparameters are available, I do have reproducibility concerns. There are many moving parts (e.g., learning the model, the constrained optimization, etc.) that to the naked eye seems sensitive to implementation details and hyperparameters.

2. Is this definition of skill commonly used? As I understand it is not. What was the reasoning behind this definition? If there is a citation for this definition of skill, that would be appreciated and helpful to the paper.

3. I am confused about this description of distance to skill: "To evaluate the ability of a policy to achieve a given skill z, we estimate the expected distance to skill, denoted d(z), with n = 10 rollouts while following skill z." What is the mathematical definition of distance to skill? Is this just Euclidean distance? If I'm missing the mathematical definition somewhere, I apologize.

4. In the experiment section, we state: "we combine with four different feature functions that we call feet contact, jump, velocity and angle." Where do these feature functions come from?

5. Where does the delta in P2 come from? When you reformulate P2 into (1), where does the delta go?

6.  In (2), which is used to train the network, delta' is needed to compute the targets for the cross-entropy loss. I checked the SCOPA hyperparameters in the Appendix and I couldn't find the values for delta'. Is the need for delta' lost somewhere?

7. Typically in goal-conditioned RL, you have a goal, and the reward function that you optimize is associated to the goal. This paper states that the P1 generalizes goal-conditioned RL, but to me it is just a different objective. I.e., you still have a main task reward augmented by the skill/goal objective. Thoughts on this? Am I misunderstanding?

---

> ### Author Response · Authors · 2023-11-20
> **Response to reviewer 1/3**
>
> First and foremost, we would like to thank you for your thorough review and insightful comments on our work. Your feedback is invaluable and we appreciate the opportunity to improve upon the points you have raised, especially about the clarity of the paper related to the definition of skills and to the method.
>
>
> > 1. Will the code be released? While the hyperparameters are available, I do have reproducibility concerns. There are many moving parts (e.g., learning the model, the constrained optimization, etc.) that to the naked eye seems sensitive to implementation details and hyperparameters.
>
> - Yes, the full code will be released in a containerised environment that will ensure that these results can easily be reproduced.
> - The code has already been submitted as supplementary material in our original submission.
> - We acknowledge that there are 5 moving parts, namely the actor, value function, successor features, Lagrangian network and world model. However, we have made no change to the components that were already present in DreamerV3 (actor, value function and world model) and the hyperparameters of the successor features are exactly the same as the value function.
> - We have no concern about convergence issues or instabilities during training as it is actually the algorithm showing the least variance across random seeds, see Figure 4 in Section 5.4 and Figure A.7 in Appendix A.1.
>
>
> > 2. Is this definition of skill commonly used? As I understand it is not. What was the reasoning behind this definition? If there is a citation for this definition of skill, that would be appreciated and helpful to the paper.
>
> - We define the skill executed by the policy as the expected features under the policy's stationary distribution. The expected features have been used before in [1, 2, 3] to characterize the behavior of the policy. However, to the best of our knowledge, we are the first to condition the policy on the expected features. Thank you for pointing that a citation was missing, and that the explanation was unclear. We have added the citation and reformulated the definition of skills. Please let us know if any point is still unclear.
> - To explain the reasoning behind this definition, it is useful to draw the analogy with GCRL. In GCRL, the objective is to achieve a desired goal $g$, i.e. to reach a state that is in the subset $\\{\left. s \in \mathcal{S} \right\vert \phi(s) = g\\}$. In contrast, our objective is to execute a desired skill $z$, i.e. to sample a trajectory that is in the subset $\\{\left. (s\_t)\_{t \geq 0} \in \mathcal{S}^\mathbb{N} \right\vert \lim\_{T \rightarrow \infty} \frac{1}{T} \sum\_{t=0}^{T-1} \phi\_t = z\\}$ (see Section 3 for a longer discussion). Thus, the skills can be seen as trajectory-level goals. Additionnaly, it has been shown that Mutual Information Maximization methods used in URL optimize the same objective (up to a constant and to a mapping between goal and skill) than GCRL [4].
> - Therefore, we argue that our definition of skill is very similar to the definition of goal used in GCRL.
> - Finally, one of our contributions is to show that our definition of skill is expressive enough to produce diverse behaviours like jumping, running in every direction, running while spinning and walking with different feet contact rates. We added a few sentences to highlight those limitations in the final paragraph.
>
> [1] Improving Generalization for Temporal Difference Learning: The Successor Representation. Dayan. 1993\
> [2] Successor Features for Transfer in Reinforcement Learning. Barreto et al. 2017\
> [3] Discovering Policies with DOMiNO: Diversity Optimization Maintaining Near Optimality. Zahavy et al. 2022\
> [4] Variational Empowerment as Representation Learning for Goal-Based Reinforcement Learning. Choi et al. 2021
>
>
> > 3. I am confused about this description of distance to skill: "To evaluate the ability of a policy to achieve a given skill z, we estimate the expected distance to skill, denoted d(z), with n = 10 rollouts while following skill z." What is the mathematical definition of distance to skill? Is this just Euclidean distance? If I'm missing the mathematical definition somewhere, I apologize.
>
> The mathematical definition of distance to skill is Euclidean distance between desired skill and observed skill. Thank you for highlighting the need for clarity regarding the distance to skill evaluation metric, we have reformulated the corresponding paragraph to include the previously missing mathematical definition.

---

> > ### Comment · Reviewer_SA5X · 2023-11-22
> > **Reviewer Response to 1/3**
> >
> > Thanks for addressing my concerns about stability and reproducibility.
> >
> > > We define the skill executed by the policy as the expected features under the policy's stationary distribution. The expected features have been used before in [1, 2, 3] to characterize the behavior of the policy. However, to the best of our knowledge, we are the first to condition the policy on the expected features. Thank you for pointing that a citation was missing, and that the explanation was unclear. We have added the citation and reformulated the definition of skills. Please let us know if any point is still unclear.
> >
> > Sure, it is natural that the expected features achieve by a policy says something about its behavior (e.g., as in the papers you cited). It seems more related to inverse RL where methods try to match demonstrator features. In this case you try to match some expected feature vector, so in that sense I see the similarity.
> >
> >
> > > To explain the reasoning behind this definition, it is useful to draw the analogy with GCRL.... Therefore, we argue that our definition of skill is very similar to the definition of goal used in GCRL."
> >
> > As per my Q7, this analogy doesn't seem to hold very strongly. I agree there is some similarity in flavor, but I still think there is a difference. In particular, in response to
> > > In GCRL, the objective is to achieve a desired goal , i.e. to reach a state that is in the subset ...
> >
> > This is "typically" the case but by no means how goal-conditioned RL is defined. In GCRL you have goals (which can have abstract representations), with associated reward functions that are a function of those goals.
> >
> > E.g., quoting
> > [1] Universal Value Function Approximators. Schaul et al. 2015
> >
> > "By universal, we mean that the value function can generalise to any goal g in a set G of possible goals: for example a discrete set of goal states; their power set; a set of continuous goal regions; or a vector representation of arbitrary pseudo-reward functions"
> >
> > SCOPA offers a solution to one specific kind of goal representation/goal space (the space of expected features) with an associated reward function (matching this  "goal" feature vector in expectation, the second term in P1). So SCOPA is a special case of GCRL, rather than a generalization. I hope this makes sense.
> >
> > > The mathematical definition of distance to skill is Euclidean distance between desired skill and observed skill. Thank you for highlighting the need for clarity regarding the distance to skill evaluation metric, we have reformulated the corresponding paragraph to include the previously missing mathematical definition.
> >
> > Thank you for this clarification. I did not see it in a revision, but I trust the authors will include this.

---

> > > ### Author Response · Authors · 2023-11-23
> > > **Response to reviewer 1/1**
> > >
> > > First and foremost, we would like to thank you for your quick reply and insightful comments on our work.
> > >
> > > > SCOPA offers a solution to one specific kind of goal representation/goal space (the space of expected features) with an associated reward function (matching this "goal" feature vector in expectation, the second term in P1). So SCOPA is a special case of GCRL, rather than a generalization. I hope this makes sense.
> > >
> > > Thank you for your insightful comments regarding our claim, we appreciate the opportunity to clarify our stance about generalizing GCRL.
> > >
> > > First, we want to say that we completely agree with this and we would like you to know that we have removed any confusing claims about generalizing GCRL. We acknowledge that GCRL methods that use state-based goals are a special case of GCRL. The idea we were trying to convey is that instead of trying to achieve a goal at state-level (like GCRL methods that use state-based goals do), SCOPA is instead trying to achieve that goal "in average" over the trajectory. We think this is a useful analogy to understand what SCOPA's skill representation looks like.
> > >
> > > We have updated the problematic paragraph at the end of Section 3, to make it clear that we draw an analogy between state-based GCRL methods and SCOPA.
> > >
> > >
> > > > Thank you for this clarification. I did not see it in a revision, but I trust the authors will include this.
> > >
> > > We now have updated Section 5.3, the definition of expected distance to skill now reads as follows:
> > > "To evaluate the ability of a policy to achieve a given skill $z$, we estimate the *expected distance to skill*, denoted $d(z)$, by averaging the euclidean distance between the desired skill $z$ and the observed skill over 10 rollouts."
> > >
> > > > Ok, this is what I was asking about. Thanks for clarifying. Updating that in the paper would be helpful as you said.
> > >
> > > We have included the two following sentences in Section 2.1:
> > > "Additionally, we focus on the case where each state $s_t$ is associated with observable features $\phi_t = \phi(s_t) \in \Phi \subset \mathbb{R}^d$. In other cases, $\phi$ can be learned with a neural network, but that is not the focus of the present work."
> > >
> > > If you think that we should highlight this fact in a different part of the paper, please let us know.
> > >
> > >
> > > Please let us know if you have any remaining concerns with the paper.

---

> ### Author Response · Authors · 2023-11-20
> **Response to reviewer 2/3**
>
> > 4. In the experiment section, we state: "we combine with four different feature functions that we call feet contact, jump, velocity and angle." Where do these feature functions come from?
>
> - In Section 5.1, we describe the tasks used in the experiments, and then in the four subsequent paragraphs we give additional information about the four feature functions and indicate where they come from, see more details below.
> - **Velocity** feature function comes from GCRL literature [1]. The skills for velocity can be directly opposite to solving the task (going forward).
> - **Angle** feature function comes from GCRL literature [1] and has been used in robotics to rotate a body in space to a target orientation. We wanted to push the boundaries of what is possible with the humanoid like moonwalking, running sideways, changing angle during rollout and so on. We believe that the dexterity coming from angle skills could definitely be useful in downstream tasks like soccer play, see bit.ly/scopa.
> - **Feet contact** feature function comes from QD literature [3] and has been extensively studied to solve downstream robotics tasks.
> - **Jump** feature function is a new feature function we introduce in this work for three reasons. First, the jump features present a significant challenge because unlike velocity, the jump features information is not directly part of the state of the agent and because the $\max$ operator makes the features function non-linear. Second, to achieve a jump skill $z = \frac{1}{T} \sum_{i=0}^{T-1} \phi_i = 10 \text{cm}$, the agent is forced to oscillate around that value because of gravity, showing the benefit to define skills as the average features instead of trying to reach the features 10cm at every time steps. Third, jumping is a useful skill for downstream tasks like jumping over hurdles.
> - The feature functions are not learned in this work, but it has been shown that it is possible [4] and would be the subject of a follow-up work. Whether it is a limitation or not depends on the need of the user of the algorithm. If the goal of the user is to learn skills like jumping, then it is very useful to be able to provide the feature function directly. However, if the goal of the user is simply to learn skills to adapt to unforeseen situations like changes in friction, then it could be desirable to learn the feature function.
>
> We have improved the clarity and readability of Section 5.1, thank you for pointing out missing information about the feature functions used in this work.
>
> [1] Goal-Conditioned Reinforcement Learning: Problems and Solutions. Liu et al. 2022\
> [2] Learning Goal-Conditioned Policies Offline with Self-Supervised Reward Shaping. Mezghani et al. 2023\
> [3] Policy Gradient Assisted MAP-Elites. Nilsson et al. 2021\
> [4] Autonomous skill discovery with Quality-Diversity and Unsupervised Descriptors. Cully et al. 2019
>
>
> > 5. Where does the delta in P2 come from? When you reformulate P2 into (1), where does the delta go?
>
> - We relax the constraint from Problem P1 using the $L^2$ norm and $\delta$, a threshold that quantifies the maximum acceptable error between the desired skill and the observed skill. The goal is to transform the "equality constraint" from P1 into an "inequality constraint" that is less restrictive and easier to satisfy. Relaxing a constraint that way is a common trick in constrained optimization. We have made adjustements in the paper to improve the clarity on this point.
> - When we reformulate P2 into Equation (1), $\delta$ goes into the Lagrange multiplier. At the end of Section 4.1, we give the objective of the Lagrange multiplier and you can see that $\delta$ appears in the expression. The general idea is that if the constraint is not satisfied (blue term greater than $\delta$), then the Lagrange multiplier will increase to put more weight on the constraint. On the contrary, if the constraint is satisfied (blue term less than $\delta$), then the Lagrange multiplier will decrease to put more weight on the objective.

---

> > ### Comment · Reviewer_SA5X · 2023-11-22
> > **Reviewer Response to 2/3**
> >
> > > The feature functions are not learned in this work, but it has been shown that it is possible [4] and would be the subject of a follow-up work.
> >
> > Ok, this is what I was asking about. Thanks for clarifying. Updating that in the paper would be helpful as you said.
> >
> > Thank you for clarifying the math in P2.

---

> ### Author Response · Authors · 2023-11-20
> **Response to reviewer 3/3**
>
> > 6. In (2), which is used to train the network, delta' is needed to compute the targets for the cross-entropy loss. I checked the SCOPA hyperparameters in the Appendix and I couldn't find the values for delta'. Is the need for delta' lost somewhere?
>
> - We have reformulated the proposition as well as Section 4.1 to improve the clarity of the paper. Specifically, we have removed $\epsilon$ and $\delta^\prime$, and thus only the hyperparameter $\delta$ remains.
> - The value of $\delta$ is task specific and you can find it in Appendix C.1.
> - The heuristic we use to choose the value of $\delta$ is the side length of the skill space divided by 100.
> - Keep in mind that $\delta$ represents the maximum acceptable error between the desired skill and the observed skill, and needs to be chosen by the user. For example, with the jump features, the skill space is $[0., 0.25]^2$ m, and we take $\delta = 0.0025 \text{m} = 2.5 \text{mm}$. With the angle features, the skill space is $]-\pi, \pi]$, and we take $\delta = 0.06$.
>
>
> > 7. Typically in goal-conditioned RL, you have a goal, and the reward function that you optimize is associated to the goal. This paper states that the P1 generalizes goal-conditioned RL, but to me it is just a different objective. I.e., you still have a main task reward augmented by the skill/goal objective. Thoughts on this? Am I misunderstanding?
>
> The statement that P1 generalizes GCRL was based on two arguments:
> - First, if we remove the (optional) objective maximization in P1 (i.e. red term in Section 4.1), then we end up only with the constraint. The constraint is responsible for executing skills, and an analogy can be drawn between "executing a skill" in our work and "reaching a goal" in GCRL. In that sense, we stated that P1 generalizes GCRL, because we can include an (optional) objective maximization in addition to "executing a skill".
> - Second, we can show that the feature space (or goal space) $\Phi$ is a subset of the skill space $\mathcal{Z}$. In that sense, we stated that P1 generalizes GCRL to trajectory-level goals. For example, the feet contact tasks illustrate that the feature space is finite equal to $\\{0, 1\\}^2$ whereas the skill space is infinite equal to $[0., 1.]^2$.
>
> However, the arguments rely on an analogy, and we agree with you that P1 is just a different objective and that it is not a generalization in any rigorous sense. Therefore, we have reformulated the corresponding sentence and paragraph at the end of Section 3. The sentence now reads as follows: "Problem P1 amounts to maximizing the return _while_ executing a desired skill. If we ignore the objective and only consider the constraint, we can show that our problem is related to GCRL."
>
>
> We once again thank you for your constructive critique. Your feedback has been instrumental in refining our paper, and we hope that our responses adequately address your concerns. We are committed to contributing valuable and clear research to the field, and your input has been essential in guiding our efforts towards this goal. Should you have any remaining doubts or concerns, please do not hesitate to share them with us. We will be more than happy to incorporate your feedback, as we did with the initial review, to further improve the quality of our work.

---

> > ### Comment · Reviewer_SA5X · 2023-11-23
> > **Review Response to Author response 3/3**
> >
> > Sorry for the late reply.
> >
> > > We have reformulated the proposition as well as Section 4.1 to improve the clarity of the paper...
> >
> > Ok, thank you, this helps. And was $\delta$ tuned?
> >
> > > However, the arguments rely on an analogy, and we agree with you that P1 is just a different objective and that it is not a generalization in any rigorous sense
> >
> > I think these issues have been discussed/clarified in our other discussions.
> >
> > > We once again thank you for your constructive critique.
> >
> > And I appreciate that you have made significant efforts to clear my confusion and improve the paper. I don't want to leave you in anticipation regarding a change of score. But as there were many changes made to the paper, I still need to reconsider the overall impact of these changes on my overall assessment of the paper. I do acknowledge that many of my concerns were addressed and questions answered. I think the paper is much better and I have a much more favorable opinion of it and will strongly consider this moving into the reviewer discussion phase.

---

> > > ### Author Response · Authors · 2023-11-23
> > > **Response to reviewer 1/1**
> > >
> > > > Ok, thank you, this helps. And was $\delta$ tuned?
> > >
> > > A general heuristic we used for $\delta$ is to take the side length of the skill space and divide by 100. The only notable exception is the Ant Feet Contact task, where the skill space is of dimension 4 which makes the task more difficult if we also divide by 100 (all the other environments have a skill space of dimension 1 or 2). For that reason, in the Ant Feet Contact task, $\delta$ is equal to the side length of the skill space divided by 10 instead of 100. You can find the values of $\delta$ in Table C.1 in appendix.
> > >
> > > $\delta$ is the equivalent of $\epsilon$ in Equation 6 in [1]. $\epsilon$ is a parameter that is commonly used in GCRL to quantify the maximum acceptable distance between the desired goal and the current achieved goal [1]. Indeed, it is generally very difficult to perfectly reach a goal $g$, so a common approach used in state-based GCRL, is to consider that a goal is achieved when $||\phi(s_{t+1}) - g|| \leq \epsilon$. $\delta$ fulfills exactly the same role as $\epsilon$ in our method.
> > >
> > > You can see why the value of $\delta$ (or $\epsilon$) depends on the task. For a task where the skills (or goals) are velocities that range between -5. and 5 m/s, a reasonable threshold would be 0.5 m/s for example. However, for a task where the skills are feet contact rates that range between 0. and 1., a reasonable threshold would be 0.1 for example. In the end, it is up to the user to decide what is a reasonable threshold for his application.
> > >
> > > [1] Goal-Conditioned Reinforcement Learning: Problems and Solutions. Liu et al. 2022

---

### Author Response · Authors · 2023-11-20
**General response to reviewers**

We would like to thank all the reviewers for their valuable and detailed feedback, which we have taken into consideration to improve our paper. We respond individually to each reviewer, addressing each point they raised.

---

### Author Response · Authors · 2023-11-23
**Final response to reviewers**

As the rebuttal is coming to an end, we would like to thank all the reviewers for their thorough reviews and insightful comments on our work. We are committed to contributing valuable and clear research to the field, and your input has been essential towards this goal, and we are confident that these revisions have significantly strengthened the paper. We hope that we have addressed all of your concerns adequately.

We have added a **latexdiff** in the supplementary materials for your convenience. This document highlights the modifications made since our initial submission.

# Improvements

We would like to highlight the major improvements we have made to the paper in response to the comments we received:
- **Readability**: We have significantly improved the clarity and readability throughout the paper. Most importantly, we have improved the problem formulation and the description of the method in Section 4.1. Specifically, we have moved the previous proposition in appendix and replaced it with a better one, that has better notations and formulation.
- **Additional Experiments**: In an effort to better support the claim that our method is able to adapt to out-of-distribution perturbations in the environments, we have added two new few-shot adaptation tasks. You can find the results in Figure 5 and you can find more information about the evaluation tasks in Appendix C.
- In Section A.3, we report all the distance/performance profiles and heatmaps without any filtering with $d_{\text{eval}}$. On those plots, all performance scores are reported, even those for which the target skill is not successfully executed.
- In Section C.2, we have added additional details about the adaptation tasks from Figure 5.
- In Section D.1, we have added more information about the baselines, especially about SMERL and how we compute the distance to skill.
- In Section F, we provide additional details regarding our evaluation metrics, specifically about the distance and performance profiles and how to read them.
- We replaced the heatmaps on Figure 3 from Humanoid Feet Contact to Ant Velocity because the figure is more interesting to comment in the Results section. All heatmaps are still available in appendix.

# Summary

In summary, we present a quality-diversity reinforcement learning algorithm, SCOPA, that learns to execute a continuous range of skills while maximizing return. The contributions are as follows:
- We derive an upper bound of the distance between expected features and skill using successor features.
- Based on this theoretical result, we extend the generalized policy iteration framework with a policy skill improvement update that is analogous to the classic policy improvement update.
- We unify value function and successor features policy iteration with constrained optimization.
- Our empirical evaluations suggest that SCOPA outperforms several skill-conditioned reinforcement learning and quality-diversity methods on continuous control locomotion tasks.
- Quantitative results demonstrate that SCOPA can extrapolate to out-of-distribution changes in the environments.
- Qualitative analyses showcase a range of remarkable behaviors (http://bit.ly/scopa).